# Theoretical refinement of CLIP by utilizing linear structure of optimal similarity

## Abstract

In this study, we propose an enhancement to the similarity computation mechanism in multi-modal contrastive pretraining frameworks such as CLIP. Prior theoretical research has demonstrated that the optimal similarity metrics between paired modalities should correspond to the pointwise mutual information (PMI) between the two modalities. However, the current implementations of CLIP and its variants fail to fully utilize the underlying linear structure of PMI. We therefore propose KME-CLIP, which leverages this structure through the inner product in a reproducing kernel Hilbert space. We theoretically prove that our method can approximate PMI with arbitrary accuracy and empirically demonstrate that our approach overall outperforms the standard CLIP formulation across several retrieval and classification tasks.

## 1 Introduction

CLIP (Radford et al., 2021) represents one of the most successful approaches in multimodal representation learning. It implements contrastive learning to align images with their corresponding captions, demonstrating exceptional performance across various tasks including retrieval (Li et al., 2022a), zero-shot image classification (Li et al., 2021), and linear classification (Jia et al., 2021). The CLIP framework has also been extended to other modality pairs such as audio and text (Guzhov et al., 2022; Elizalde et al., 2023). However, to obtain more fine-grained multimodal representations, researchers have continued to improve CLIP through modification of loss functions (Goel et al., 2022; Mu et al., 2022) or similarity functions (Desai et al., 2023; Uesaka et al., 2025).

We focus on similarity functions in contrastive learning. It is theoretically known that the similarity function minimizing the population contrastive loss is given by the pointwise mutual information (PMI), defined as $\log(p(x, y)/(p(x)p(y)))$ for two datapoints $x$ and $y$ from different modalities (Oord et al., 2018; Zhang et al., 2023). Moreover, it has been shown that, when the similarity function closely approximates PMI, its performance improves across various downstream tasks including zero-shot (Oko et al., 2025) and linear classification (Uesaka et al., 2025). In this paper, we identify that, under a certain condition, the exponential of PMI possesses a linear structure of an inner product in $L^2$ space, which remains unexploited in the similarity computation of CLIP and its modifications; ignoring it leads to failure in a toy example.

To exploit this structure, we propose KME-CLIP (Kernel Mean Embedding CLIP), which projects embeddings into a reproducing kernel Hilbert space (RKHS) where the linear structure of inner products of infinite dimensional vectors in $L^2$ space can be naturally captured through inner products. Specifically, our method represents embeddings from image and text encoders as positive functions in an RKHS and computes the logarithm of their inner product as a similarity metric, thereby enabling more effective PMI approximation.

Our contributions are summarized as follows:

1. We identify that PMI exhibits an inherent linear structure under reasonable assumptions regarding data distributions (Section 3.1), which remains unexploited in conventional CLIP. We then propose KME-CLIP, whose similarity leverages this linearity through RKHS and harnesses the functional representation capacity of reproducing kernels (Section 3.2).

2. We theoretically establish that the similarity metric in KME-CLIP can approximate PMI with arbitrary accuracy when the size of point set is sufficiently increased (Section 4.1).

Furthermore, we demonstrate theoretically that CLIP faces inherent limitations in approximating PMI under certain conditions (Section 4.2).

3. In our experiments (Section 5), we train models on CC3M (Sharma et al., 2018) and CC12M (Changpinyo et al., 2021) datasets, and demonstrate that KME-CLIP consistently outperforms CLIP on several retrieval tasks using CC3M, MSCOCO (Chen et al., 2015), and Flickr30K (Plummer et al., 2015). We further evaluate zero-shot and linear classification accuracy across more than 10 diverse datasets including ImageNet (Russakovsky et al., 2015) and CIFAR-100 (Krizhevsky, 2009), where KME-CLIP exhibits superior performance on most of the tasks.

## 2 Preliminaries

### 2.1 Contrastive language-image pre-training (CLIP)

CLIP (Radford et al., 2021) trains image encoder and text encoder to maximize the cosine similarity between embeddings of the corresponding image and caption pairs. This method aims to learn the representation which projects the paired image and caption to aligned embedding vectors.

While we refer to them as image and text for simplicity, our arguments hold for any pair of modalities. Let $g^{\mathcal{X}} : \mathcal{X} \to \mathbb{R}^d$ and $g^{\mathcal{Y}} : \mathcal{Y} \to \mathbb{R}^d$ be $L^2$-normalized image and text encoders, respectively. Given a minibatch of image dataset and the corresponding caption dataset $\{(x_i, y_i)\}_{i=1}^n$, the training loss, which we call the CLIP loss, is calculated as follows:

$$\frac{1}{2}\left[-\frac{1}{n}\sum_{i=1}^n \log\left(\frac{\exp(\frac{1}{\tau}g^{\mathcal{X}}(x_i)^\top g^{\mathcal{Y}}(y_i))}{\sum_{j=1}^n \exp(\frac{1}{\tau}g^{\mathcal{X}}(x_j)^\top g^{\mathcal{Y}}(y_i))}\right) - \frac{1}{n}\sum_{i=1}^n \log\left(\frac{\exp(\frac{1}{\tau}g^{\mathcal{X}}(x_i)^\top g^{\mathcal{Y}}(y_i))}{\sum_{j=1}^n \exp(\frac{1}{\tau}g^{\mathcal{X}}(x_i)^\top g^{\mathcal{Y}}(y_j))}\right)\right].$$

Here, $\tau$ denotes the temperature parameter, which we consider to be a learnable parameter in this paper, controlling the influence of inner product magnitudes on the loss function.

The CLIP loss is a proxy of the performance on the image-to-text and text-to-image retrieval tasks: for instance, the first term is measuring the probability that we retrieve $y_i$ among $\{y_j\}_{j=1}^n$ based on the distribution induced from the logits $\{g^{\mathcal{X}}(x_i)^\top g^{\mathcal{Y}}(y_j)/\tau\}_{j=1}^n$. Thus, the minimization of the training loss implies high performance in cross-modal retrieval tasks, which is one of the primary tasks where CLIP embeddings are used. Indeed, theoretical analysis demonstrates that when the CLIP loss approaches its minimum, CLIP's performance on zero-shot classification tasks improves significantly (Oko et al., 2025).

### 2.2 Pointwise mutual information as best similarity metric

In this section, we consider a population loss variant of the CLIP loss. Let $X$ and $Y$ be $\mathcal{X}$- and $\mathcal{Y}$-valued random variables, representing image and text, respectively. Throughout our analysis, we assume the existence of density functions for both $X$ and $Y$.

**Assumption 1** (Density function). *We assume that $X$ and $Y$ have probability marginal density functions $p(x)$ and $p(y)$ as well as a joint density $p(x,y)$ with respect to reference measures $\nu_X$, $\nu_Y$, and $\nu_X \otimes \nu_Y$, respectively.*

If we generalize from the scaled similarity of CLIP $g^{\mathcal{X}}(x)^\top g^{\mathcal{Y}}(y)/\tau$ to any similarity metric $S(x,y)$, the training loss (minus $\log n$) converges to the following as we increase the minibatch size $n$:

$$L_S := \frac{1}{2}\mathbb{E}_{p(x,y)}\left[-\log\frac{\exp(S(x,y))}{\mathbb{E}_{p(x')}[\exp(S(x',y))]}\right] + \frac{1}{2}\mathbb{E}_{p(x,y)}\left[-\log\frac{\exp(S(x,y))}{\mathbb{E}_{p(y')}[\exp(S(x,y'))]}\right]. \quad (1)$$

We note that minimizing $L_S$ directly enhances CLIP's performance in image-to-text and text-to-image retrieval tasks, analogous to the effect of minimizing the standard CLIP loss.

Remarkably, the optimal solution $S(x,y)$ of $L_S$ is known theoretically. We first define pointwise mutual information as the probabilistic relationship between two modalities.

**Definition 1** (Pointwise mutual information). *For $X, Y$ under Assumption 1, the pointwise mutual information* (PMI) *between them at points $x \in \mathcal{X}$ and $y \in \mathcal{Y}$ is defined as $\mathrm{PMI}(x,y) := \log \frac{p(x,y)}{p(x)p(y)}$.*

We now establish that the optimal similarity function $S(x, y)$ that minimizes $L_S$ corresponds to PMI, as formalized in the following theorem.

**Theorem 1** (Proposition1, Zhang et al. (2023)). *$L_S$ in (1) attains its minimum if we have $S(x, y) = \text{PMI}(x, y) + \text{const}$ for all $x \in \mathcal{X}$ and $y \in \mathcal{Y}$.*

It is also known that the minimum of $L_S$ is $I(X, Y)$, the mutual information between $X$ and $Y$. From Theorem 1, constructing a similarity function that aligns with PMI is an important direction for improving the accuracy of contrastive learning. Motivated by this, we investigate what kinds of similarity functions can satisfy this requirement in the next section.

## 3 PROPOSED METHOD

### 3.1 KEY INSIGHT: LINEAR STRUCTURE IN $\exp(\text{PMI})$

Our key insight is that $\exp(\text{PMI})$ can be expressed as an inner product in $L^2$ space under certain conditions, where $\exp(\text{PMI})$ simply refers to the exponential of PMI from Definition 1. To formalize this insight, we introduce the following assumption.

**Assumption 2** (Conditional independence). *We assume that there exists a latent variable $Z$ taking values in a compact metric space $\mathcal{Z}$, with a density function $\rho(z)$ with respect to a reference (Borel) measure $\nu$. We further assume that the conditional densities satisfy $p(x, y|z) = p(x|z)p(y|z)$.*

Assumption 2 is about conditional independence, which is often adopted in the theoretical study of multi-modal learning (Chen et al., 2024; Oko et al., 2025). Intuitively, this assumes the existence of a latent variable that represents the underlying topic of images and texts.

Under this assumption, $\exp(\text{PMI})$ is equivalent to the inner product in an $L^2$ space:

$$\frac{p(x, y)}{p(x)p(y)} = \frac{1}{p(x)p(y)} \int p(x|z)p(y|z) \, \mathrm{d}\rho(z) = \left\langle \frac{p(x|\cdot)}{p(x)}, \frac{p(y|\cdot)}{p(y)} \right\rangle_{L^2(\rho)},$$

where $\mathrm{d}\rho(z) = \rho(z)\mathrm{d}\nu(z)$ gives a probability measure over $\mathcal{Z}$.

Thus, a straightforward way to interpret $\exp(\text{PMI})$ is that it is an inner product between two embedding functions in the $L^2$ space. However, CLIP tries to approximate this "linear" object by $\exp(g^{\mathcal{X}}(x)^{\top} g^{\mathcal{Y}}(y)/\tau)$, which is inherently nonlinear due to the exponential operation.

To exploit the linearity structure of $\exp(\text{PMI})$, we consider projecting the embedding vectors onto an RKHS, which is an infinite dimensional linear space, and defining the logarithm of the inner product in RKHS as the similarity metric corresponding to $S(x, y)$.

### 3.2 PROPOSED METHOD: KME-CLIP

From the discussion in the previous section, we propose *kernel mean embedding CLIP* (KME-CLIP), which approximates the $L^2$ inner product by an inner product in the RKHS (see also Section 3.3 for the implementation). Let $k$ be a symmetric positive definite kernel on $\mathbb{R}^d$ and $\mathcal{H}$ be the associated RKHS. We further assume the positivity of the kernel ($k > 0$) for implementation reasons. The image embedding function of KME-CLIP has the following components ($\mathbb{R}_+$ is the set of positive real numbers):

- Encoders: $f_i^{\mathcal{X}} : \mathcal{X} \to \mathbb{R}^d$ for $i = 1, \dots, m_{\mathcal{X}}$.
- Positive weight functions: $w_i^{\mathcal{X}} : \mathcal{X} \to \mathbb{R}_+$ for $i = 1, \dots, m_{\mathcal{X}}$.

We refer to the multiple embeddings generated from the several encoders $f_i^{\mathcal{X}}$ as a *point set* and define the number of encoders, $m_{\mathcal{X}}$, as its size. These encoders and weight functions are learnable, while the size of point set $m_{\mathcal{X}}$ is fixed as a hyper-parameter. Similarly, KME-CLIP has their text-side counterpart: $m_{\mathcal{Y}}$, $f_j^{\mathcal{Y}}$, and $w_j^{\mathcal{Y}}$. Given the kernel $k$, RKHS $\mathcal{H}$, and these components, we embed $x \in \mathcal{X}$ and $y \in \mathcal{Y}$ into the RKHS as follows:

$$h_{\theta}^{\mathcal{X}}(x) := \sum_{i=1}^{m_{\mathcal{X}}} w_i^{\mathcal{X}}(x)k(f_i^{\mathcal{X}}(x), \cdot) \in \mathcal{H}, \qquad h_{\theta}^{\mathcal{Y}}(y) := \sum_{j=1}^{m_{\mathcal{Y}}} w_j^{\mathcal{Y}}(y)k(\cdot, f_j^{\mathcal{Y}}(y)) \in \mathcal{H}. \qquad (2)$$

Then we define the logarithm of their inner product in $\mathcal{H}$ as our proposed similarity metric:

$$S(x,y) = \log \left\langle h_\theta^{\mathcal{X}}(x), h_\theta^{\mathcal{Y}}(y) \right\rangle_{\mathcal{H}} = \log \sum_{i=1}^{m_{\mathcal{X}}} \sum_{j=1}^{m_{\mathcal{Y}}} w_i^{\mathcal{X}}(x) w_j^{\mathcal{Y}}(y) k(f_i^{\mathcal{X}}(x), f_j^{\mathcal{Y}}(y)), \tag{3}$$

where $\langle \cdot, \cdot \rangle_{\mathcal{H}}$ denotes the inner product in $\mathcal{H}$. Since the inside of the logarithm must be positive, we have constrained $k$ and weight functions to be positive. Because of the positivity of weight functions, $\mu_x := \sum_{i=1}^{m_{\mathcal{X}}} w_i^{\mathcal{X}}(x) \delta_{f_i^{\mathcal{X}}(x)}$, where $\delta_a$ is the delta distribution at point $a \in \mathbb{R}^d$, becomes a positive measure. Then, the embedding in (2) is rewritten as $x \mapsto \int k(z, \cdot) \, \mathrm{d}\mu_x(z)$, which is called kernel mean embedding when $\mu_x$ is a probability measure (Muandet et al., 2017). This is why our method is named KME-CLIP.

By choosing an appropriate kernel $k$, we can prove that the similarity metric with the form (3) is capable of approximating PMI with arbitrary accuracy as we increase $m_{\mathcal{X}}$ and $m_{\mathcal{Y}}$ (see Section 4).

### 3.3 IMPLEMENTATION

To generate multiple point set image embeddings $\{f_i^{\mathcal{X}}(x)\}_{i=1}^{m_{\mathcal{X}}}$, we leverage the intermediate features from Vision Transformer (Dosovitskiy et al., 2021), which naturally produces multiple token outputs. To derive the positive weights $\{w_i^{\mathcal{X}}(x)\}_{i=1}^{m_{\mathcal{X}}}$, we similarly utilize these intermediate features by mapping them to additional features and applying an activation function to ensure positivity. In the same way as CLIP, we apply $L^2$-normalization for each image embedding $f_i^{\mathcal{X}}(x)$.

For the text embeddings, we utilize multiple output tokens from the Transformer architecture (Vaswani et al., 2017). Regarding the text positive weights $\{w_i^{\mathcal{Y}}(y)\}_{i=1}^{m_{\mathcal{Y}}}$, we process the Transformer outputs using the same methodology applied to image embeddings.

Figure 1 shows a summary of our proposed method. A comparable approach is implemented in WPSE (Uesaka et al., 2025); readers may refer to Figure 2 therein for a more detailed illustrative example.

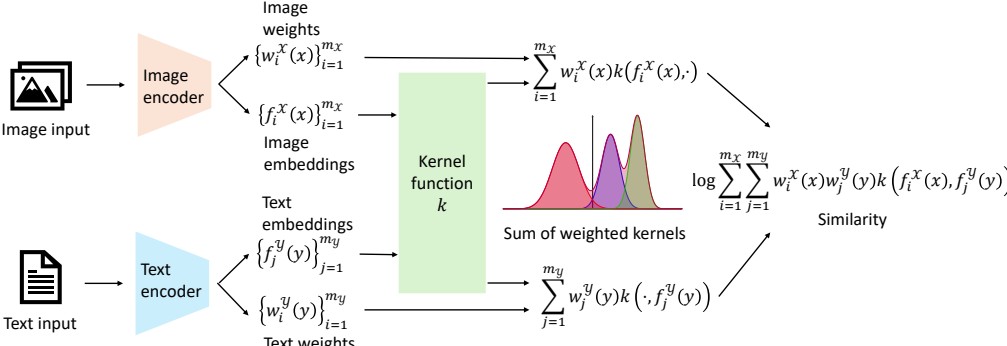

Figure 1: Overview of our proposed method. We project the image and text embedding into RKHS and calculate similarity by the logarithm of the inner product in this space.

Though our method is available by any positive kernel $k$, we primarily focus on the case of the Gaussian kernel. This is from two reasons. First is the capability of taking the value near zero. This is for approximating a situation where $p(x, y) = 0$ holds. Such situations occur when the image and provided caption do not correspond at all, such as an image of a dog and a text *this is a cat*. Second is since by Gaussian kernels, our method involves CLIP as a special case, as is stated later in Section 3.4.1. This property provides a certain degree of assurance that our method will achieve performance at least comparable to that of CLIP.

### 3.4 COMPARISON WITH OTHER EXISTING METHODS

In this section, we compare KME-CLIP with the two most relevant methods, CLIP and WPSE (Uesaka et al., 2025). For the full exposition of relevant literature, see Section 6.

### 3.4.1 COMPARISON WITH CLIP

CLIP uses the form of $g^{\mathcal{X}}(x)^\top g^{\mathcal{Y}}(y)/\tau$ as the similarity metric between the embedding vectors $g^{\mathcal{X}}(x)$ and $g^{\mathcal{Y}}(y)$. CLIP can be regarded as a special case of KME-CLIP, when $g^{\mathcal{X}}$ and $g^{\mathcal{Y}}$ are normalized vectors in $\mathbb{R}^d$. Consider using the Gaussian kernel $k = k_\sigma$, *i.e.*, $k_\sigma(u, v) := \exp(-\|u - v\|^2/2\sigma^2)$. If we let $m_{\mathcal{X}} = m_{\mathcal{Y}} = 1$, $w_1^{\mathcal{X}}(x) = w_1^{\mathcal{Y}}(y) = 1$, and $\sigma^2 = \tau$, then we have the following proposition:

**Proposition 2.** *Under the setting above, the similarity function of KME-CLIP corresponds to that of CLIP excluding the addition of a constant.*

The proof is in Appendix B. We remark that the constant factor in the statement does not affect the property of the model since it vanishes in the loss calculation. From Proposition 2, the similarity function of KME-CLIP is at least as expressive as CLIP if we use the same Euclidean embedding space $\mathbb{R}^d$. It also represents a practical generalization of CLIP by incorporating learnable weights and utilizing larger size of point set in its implementation. Moreover, while KME-CLIP apparently uses more embedding parameters than CLIP, we can just reuse the discarded outputs of CLIP's transformer implementation, thus the network size for obtaining CLIP and KME-CLIP is almost identical.

While CLIP can be viewed as a one-point version of our method, their theoretical properties differ fundamentally: our method leverages a linearity structure, whereas CLIP does not. The logarithmic term of our similarity function, $\langle h_\theta^{\mathcal{X}}(x), h_\theta^{\mathcal{Y}}(y)\rangle_{\mathcal{H}}$, exhibits a bi-linear structure with respect to functions of $x$ and $y$. In contrast, the logarithmic term of CLIP's similarity function, $\exp(g^{\mathcal{X}}(x)^\top g^{\mathcal{Y}}(y)/\tau)$, does not possess such a bi-linear structure due to the non-linear transformation introduced by the $\exp$ function. This distinction implies that, even in settings closer to real-world data, CLIP encounters difficulties in approximating PMI, whereas our method is able to approximate it effectively, as demonstrated later in Theorems 6 and 7.

### 3.4.2 COMPARISON WITH WPSE

WPSE (Uesaka et al., 2025) embeds datapoints into an RKHS in the same way as our method, and it calculates similarity in the form of $\langle u, v\rangle_{\mathcal{H}}/\tau$ without $\log$ to approximate PMI:

$$S(x, y) = \sum_{i,j} w_i'^{\mathcal{X}}(x) w_j'^{\mathcal{Y}}(y) k(f_i'^{\mathcal{X}}(x), f_j'^{\mathcal{Y}}(y)),$$

where $f_i'^{\mathcal{X}}$ and $f_i'^{\mathcal{Y}}$ are image and text encoders, $w_i'^{\mathcal{X}}$ and $w_i'^{\mathcal{Y}}$ are image and text weights. They also utilize multiple outputs from the Transformer architecture to obtain $f_i'^{\mathcal{X}}$ and $f_i'^{\mathcal{Y}}$. The implementation methodology for generating embeddings and weights is identical to our approach, with the only difference being the specific activation function used to ensure weight positivity in ours.

However, the underlying motivation of WPSE is fundamentally different from KME-CLIP. WPSE is motivated by transforming CLIP embeddings into infinite-dimensional vectors from the perspective of representational capacity, whereas our approach aims at leveraging inherent linear structure in $\exp(\mathrm{PMI})$, which is ignored both by CLIP and WPSE. This distinction results in differences both in the theoretical properties discussed in the following remark and in the empirical performance reported later in Section 5.

**Remark 1.** Although Uesaka et al. (2025) also showed that WPSE can approximate PMI with small errors under the assumption that the PMI function is Lipschitz continuous, this assumption may be violated when the joint distribution contains a pair $(x, y)$ such that $p(x, y) = 0$ while $p(x), p(y) > 0$, because $\log \frac{p(x,y)}{p(x)p(y)}$ diverges to $-\infty$. In contrast, our theoretical analysis assumes Lipschitz continuity of $\exp(\mathrm{PMI})$, rather than of PMI itself, as stated in Assumption 3. Consequently, our framework can accommodate cases in which PMI diverges to $-\infty$.

## 4 THEORY

### 4.1 THEORETICAL GUARANTEE FOR KME-CLIP

In this section, we theoretically guarantee the performance of our method by proving the intuition proposed in Section 3.1.

First, we investigate how the difference between $\exp(\text{PMI})$ and $\exp(S(x, y))$ affects the deviation of the loss from its minimum value.

**Theorem 3.** *Let $0 < \delta \le \epsilon \le 1/e^2$. Under Assumption 1, let a function $h : \mathcal{X} \times \mathcal{Y} \to \mathbb{R}_+$ satisfy $h \ge \delta$ and $|h(x, y) - \exp(\text{PMI}(x, y))| \le \epsilon$ for all $x \in \mathcal{X}$ and $y \in \mathcal{Y}$. Then, we have*

$$L_{\log h} \le \inf_S L_S + 3\sqrt{\epsilon} + \sqrt{\epsilon} \log\left(\frac{1}{\epsilon} + \frac{1}{\delta}\right), \tag{4}$$

*where $L_S$ is the population loss of CLIP for a similarity function $S$ defined in* (1).

The proof is given in Appendix C.1. This theorem states that if the inner product of RKHS in our method approximates $\exp(\text{PMI})$ (we later show it in section 4.1), then the population loss $L_S$ is also near to its minimum. Here, $h \ge \delta$ is a technical assumption, but we can construct an example of $\delta = \epsilon$ from a function $\tilde{h}$ with $|\tilde{h} - \exp(\text{PMI}(x, y))| \le \epsilon$ by simply taking $h(x, y) = \max\{\epsilon, \tilde{h}(x, y)\}$.

Second, we show that $\exp(\text{PMI})$ can well be approximated by the integral form of the kernel mean embedding, using positive weight functions. Before stating the result, we pose an additional assumption on the regularity of the conditional densities.

**Assumption 3** (Regularity of conditional densities). *We assume that there exists a universal constant $C_L > 0$ such that, for every $x \in \mathcal{X}$ and $y \in \mathcal{Y}$, $p(x|z)/p(x)$ and $p(y|z)/p(y)$ are $C_L$-Lipschitz continuous with respect to $z$.*

With this assumption, we have the following result. We defer its proof to Appendix C.2.

**Theorem 4.** *Suppose that Assumptions 1–3 hold. Let $k$ be a Gaussian kernel on $\mathbb{R}^d$ with length-scale $\sigma$, i.e., $k(u, v) = \exp(-\|u - v\|^2/(2\sigma^2))$. For an arbitrary $\epsilon > 0$, there exist an appropriate choice of $\sigma$, a probability measure $\mu$ on $\mathbb{R}^d$, and **positive** functions $g_X^x, g_Y^y \in L^2(\mu)$ for all $x \in \mathcal{X}$ and $y \in \mathcal{Y}$ such that the following holds: For each $x \in \mathcal{X}$ and $y \in \mathcal{Y}$, we have*

$$\left| \left\langle \int_{\mathbb{R}^d} k(\cdot, u) g_X^x(u) \, \mathrm{d}\mu(u), \int_{\mathbb{R}^d} k(\cdot, u) g_Y^y(u) \, \mathrm{d}\mu(u) \right\rangle_{\mathcal{H}} - \exp(\text{PMI}(x, y)) \right| < \epsilon.$$

While there is no constraint on $\mu$, from the proof, we can additionally take $\mu$ to be a measure over the unit sphere of $\mathbb{R}^d$ (if $d \ge 2$), following the convention of CLIP.

Finally, we show the following discretization result since it is difficult to express the integrations in a practical implementation. While it essentially follows from the previous results on positively-weighted kernel quadrature (Chen et al., 2010; Bach et al., 2012; Hayakawa et al., 2022), we prove it in Appendix C.3 for completeness.

**Theorem 5.** *Let $k$ be a positive definite kernel on $\mathbb{R}^d$ such that $K := \sup_{u \in \mathbb{R}^d} k(u, u) < \infty$. Given a probability measure $\mu$ over $\mathbb{R}^d$ and $g \in L^2(\mu)$, for each positive integer $m$, there exists $u_1, ..., u_m \in \mathbb{R}^d$ such that*

$$\left\| \int_{\mathbb{R}^d} k(u, \cdot) g(u) \, \mathrm{d}\mu(u) - \frac{1}{m} \sum_{i=1}^m k(u_i, \cdot) g(u_i) \right\|_{\mathcal{H}} \le \frac{\sqrt{K} \|g\|_{L^2(\mu)}}{\sqrt{m}},$$

*where $\|\cdot\|_{\mathcal{H}}$ denotes the norm in $\mathcal{H}$.*

As this theorem states, the approximation error of the discretization decays by $O(1/\sqrt{m})$, where $m$ is the size of the point set used in the embedding (see, e.g., (2)). Note that we can take $K = 1$ as long as we use Gaussian kernels.

By combining Theorems 3 to 5, we demonstrate that the population loss in (1) can be brought arbitrarily close to its theoretical optimal value using the KME-CLIP similarity function when the size of point set is sufficiently increased. Furthermore, these results can avoid the strong assumptions such as the Lipschitz continuity of PMI as discussed in Section 3.4.2, thanks to the linearity of the inner product in RKHS.

## 4.2 THEORETICAL COMPARISON OF CLIP AND KME-CLIP UNDER LOW MODALITY GAP

In this section, we introduce a toy example that illustrates a scenario where CLIP encounters significant difficulty in approximating PMI through its similarity function, while KME-CLIP does not. Although this represents a simplified model, it effectively demonstrates the fundamental limitations of CLIP and the inherent advantages of our proposed method.

Let us naturally define in-modal similarities defined by CLIP and KME-CLIP: for $x, x' \in \mathcal{X}$, we define $S^{\mathcal{X}}(x, x') = g^{\mathcal{X}}(x)^\top g^{\mathcal{X}}(x')/\tau$ when using CLIP and $S^{\mathcal{X}}(x, x') = \log \left\langle h_\theta^{\mathcal{X}}(x), h_\theta^{\mathcal{X}}(x') \right\rangle_{\mathcal{H}}$ when the model is KME-CLIP. For readability, under Assumption 2, we specifically write $p(x)$ as $p_X(x)$ and $p(x|z)$ as $p_{X|Z}(x|z)$ in this section (same for $Y$). Let us also define $p_X(x, x') := \int p_{X|Z}(x|z) p_{X|Z}(x'|z) \, d\rho(z)$. With these notations, we can also define the in-modal PMI as $\text{PMI}^{\mathcal{X}}(x, x') := \log \frac{p_X(x, x')}{p_X(x) p_X(x')}$.

Now we introduce an interpretation of a model with low modality gap:

**Definition 2** (Low modality gap). There exists $\delta > 0$ such that the similarities $S$ and $S^{\mathcal{X}}$ satisfy, for any $x, x' \in \mathcal{X}$ with $\text{PMI}^{\mathcal{X}}(x, x') < \log \delta$, there exists $y \in \mathcal{Y}$ such that $S^{\mathcal{X}}(x, x') \leq S(x, y)$.

This property keeps the model (either CLIP or KME-CLIP) from embedding different modalities to regions completely separated to each other, preventing a large modality gap as discussed by Liang et al. (2022). Low modality gap is a desirable property for embeddings since large modality gap leads to a degradation in model performance. Incorporating in-modal self-supervised learning like SimCLR (Chen et al., 2020) into multimodal representation learning is also done by Mu et al. (2022).

Next, let us introduce the data-generation process on which we compare CLIP and KME-CLIP.

**Assumption 4** (2-mixture model). *Under Assumptions 1 &2, the latent and state spaces are given by $\mathcal{Z} = \{1, \ldots, N\}$ and $\mathcal{X} = \mathcal{Y} = \{(i, j) \mid i, j \in \mathcal{Z}\}$ with **unordered pairing**, i.e., $(i, j) = (j, i)$. The distribution of $Z$, $\rho$ is given by the uniform distribution over $\mathcal{Z}$, and for each $i, j, \ell \in \mathcal{Z}$, we let $p_{X|Z}((i, j) \mid \ell) = p_{Y|Z}((i, j) \mid \ell) = \frac{1}{N+1}(\delta^{i\ell} + \delta^{j\ell})$, where $\delta^{i\ell}$ and $\delta^{j\ell}$ are Kronecker's delta.*

This setting represents the situation like this: There exist classes $1, \ldots, N$ and for each class $i$, there exists a datapoint $(i, i)$ representing purely this class. The other datapoints, $(i, j)$ for $i \neq j$, represent a half-and-half mixture of classes $i$ and $j$. Under this setting, the following theorem states that CLIP has a difficulty in expressing $\exp(\text{PMI})$ when $N$ is large, even if we allow the degree of freedom regarding the constant term in Theorem 1. We defer its proof to Appendix C.4.

**Theorem 6.** *Under Assumption 4, let us also assume $N > 9^d$, where the CLIP embeddings $g^{\mathcal{X}}(x)$, $g^{\mathcal{Y}}(y)$ are on the unit sphere of $\mathbb{R}^d$. Then, at least one of the followings holds.*

> *1. CLIP embeddings does not satisfy the low modality gap condition.*

> *2. For any choice of constants $\alpha, \tau > 0$, there exist $x \in \mathcal{X}$ and $y \in \mathcal{Y}$ such that $\left| \alpha \exp \left( g^{\mathcal{X}}(x)^\top g^{\mathcal{Y}}(y)/\tau \right) - \exp(\text{PMI}(x, y)) \right| \geq N/4$.*

In contrast, our method can exploit the sparse structure of the data distribution as follows. We give its proof in Appendix C.5.

**Theorem 7.** *Under the same setting as Theorem 6, let us use Gaussian kernel on $\mathbb{R}^d$ ($d \geq 2$) for KME-CLIP with $m_{\mathcal{X}} = m_{\mathcal{Y}} = 2$. For any $\epsilon > 0$, there exists appropriate choice of the length-scale $\sigma$, encoders, and weight functions that satisfy the followings at the same time.*

> *1. These KME-CLIP embeddings satisfy a low modality gap condition.*

> *2. For each $x \in \mathcal{X}$ and $y \in \mathcal{Y}$, we have*
> $$\left| \left\langle h_\theta^{\mathcal{X}}(x), h_\theta^{\mathcal{Y}}(y) \right\rangle_{\mathcal{H}} - \exp(\text{PMI}(x, y)) \right| < \epsilon,$$
> *where $h_\theta^{\mathcal{X}}$ and $h_\theta^{\mathcal{Y}}$ are the embeddings given by (2).*

## 5 EXPERIMENTS

In this section, we empirically demonstrate the effectiveness of our method across several downstream tasks. Below, we describe the outline of the experimental setup.

For pre-training, we utilized the training splits of Conceptual Captions 3M (CC3M) (Sharma et al., 2018) and Conceptual Captions 12M (CC12M) (Changpinyo et al., 2021) datasets. We used the same dataset for training CLIP, WPSE, and KME-CLIP for a fair comparison. We adopted a non-pretrained ViT-B/16 encoder (Dosovitskiy et al., 2021) as the base architecture for the image encoder and a non-pretrained Transformer encoder (Vaswani et al., 2017) with width $512$, 12 layers, and 8 attention heads for the text encoder across CLIP, WPSE, and KME-CLIP. For optimization, we employed AdamW with a cosine learning rate scheduler incorporating linear warm-up, using a batch size of $1024$. The hyperparameter configurations followed those established in Uesaka et al. (2025), with consistent settings maintained across CLIP, WPSE, and KME-CLIP implementations to ensure fair comparison.

As for the implementation of KME-CLIP, we employed the Gaussian kernel for the non-linear kernel function $k$, *i.e.*, $k(u, v) := \exp(-\|u, v\|^2/2\sigma^2)$. To avoid numerical instability when $\sigma$ approaches zero, we reparameterized using $\tau = 1/\sigma^2$ and established $\tau$ as a learnable parameter in our implementation. We adapted a softplus function, *i.e.*, $\frac{1}{1+e^{-x}}$ for the activation function for the positivity. For implementation details regarding CLIP, WPSE and KME-CLIP as well as comprehensive hyperparameter specifications, refer to Appendix A.1.

We evaluate performance on retrieval, zero-shot classification, and linear classification—all standard downstream tasks for assessing multimodal representation learning capabilities. We defer detailed information regarding linear classification to Appendix A.2.

## 5.1 RETRIEVAL

We evaluated the retrieval ability of the embeddings, by the validation splits of CC3M, MSCOCO (Chen et al., 2015), and Flickr30K (Plummer et al., 2015). The choice of dataset is following previous works like Yao et al. (2022) or Desai et al. (2023). The (image-to-)text and (text-to-)image retrieval tasks we employ here are the primary target of minimizing CLIP-like losses.

We show the result in Table 1. Overall, our proposed approach demonstrates superior performance compared to the other two methods. This result reflects the superiority of KME-CLIP for approximating PMI, as discussed theoretically in Section 4.

Table 1: Text retrieval results (top) and image retrieval results (bottom). We report the top-1, 5, 10 accuracy (%).

| | | CC3M | | | MSCOCO | | | Flickr30K | | |
|---|---|---|---|---|---|---|---|---|---|---|
| | Model | Top1 | Top5 | Top10 | Top1 | Top5 | Top10 | Top1 | Top5 | Top10 |
| | CLIP | 21.95 | 41.97 | 51.01 | 13.64 | 32.34 | 43.57 | 26.46 | 52.18 | 62.68 |
| CC3M | WPSE | 21.39 | 41.36 | 50.51 | **14.78** | 33.75 | 44.86 | 27.22 | 52.80 | 63.92 |
| | KME-CLIP | **24.22** | **44.85** | **53.51** | 14.14 | **34.25** | **45.65** | **28.18** | **55.72** | **65.92** |
| | CLIP | 22.82 | 42.89 | 52.22 | **24.56** | 48.25 | 60.15 | 44.76 | 73.30 | 81.94 |
| CC12M | WPSE | 22.59 | 42.70 | 51.94 | 24.29 | 48.85 | 61.07 | 45.80 | 73.74 | 82.40 |
| | KME-CLIP | **23.36** | **44.38** | **54.19** | 23.54 | **48.87** | **61.24** | **46.80** | **75.90** | **84.68** |

| | | CC3M | | | MSCOCO | | | Flickr30K | | |
|---|---|---|---|---|---|---|---|---|---|---|
| | Model | Top1 | Top5 | Top10 | Top1 | Top5 | Top10 | Top1 | Top5 | Top10 |
| | CLIP | 25.36 | 46.01 | 54.91 | 14.57 | 34.52 | 45.29 | 27.68 | 53.24 | 64.42 |
| CC3M | WPSE | 25.36 | 46.63 | 56.05 | 15.36 | 35.29 | 46.48 | 28.48 | 54.14 | 64.64 |
| | KME-CLIP | **26.90** | **49.05** | **58.09** | **15.74** | **35.94** | **47.25** | **30.52** | **57.56** | **68.42** |
| | CLIP | 22.71 | 43.37 | 52.55 | 24.74 | 48.93 | 60.36 | 46.10 | 73.92 | 82.24 |
| CC12M | WPSE | **23.76** | 44.33 | 53.85 | 24.73 | 49.45 | 61.46 | 47.12 | 74.96 | 83.04 |
| | KME-CLIP | 23.70 | **45.51** | **55.14** | **24.93** | **50.47** | **62.53** | **47.92** | **76.82** | **85.16** |

## 5.2 ZERO-SHOT CLASSIFICATION

Zero-shot classification is an approach for image classification where, given a list of class names, the method classifies images by performing image-to-text retrieval using captions generated from these class names. We evaluated it by the test splits (for datasets without a test split, the validation split was used) of the following 13 benchmark datasets: ImageNet (Russakovsky et al., 2015), CIFAR-

10 (Krizhevsky, 2009), CIFAR-100 (Krizhevsky, 2009), STL-10 (Coates et al., 2011), Food-101 (Bossard et al., 2014), Caltech-101 (Fei-Fei et al., 2006), Stanford Cars (Krause et al., 2013), FGVC Aircraft (Maji et al., 2013), Oxford Flowers (Nilsback & Zisserman, 2008), EuroSAT (Helber et al., 2019), Describable Textures Dataset (DTD) (Cimpoi et al., 2014), Oxford Pets (Parkhi et al., 2012), and SUN397 (Xiao et al., 2010). Following SLIP (Mu et al., 2022), we adopted prompt ensembling and utilized prompts provided by SLIP for each dataset.

We show the result in Table 2. Our proposed method outperformed the other two methods.

Table 2: Zero-shot classification results. Mean per-class accuracy (%) is reported for Caltech-101, Aircraft, Flowers, and Pets datasets. Top-1 accuracy (%) is used for all other datasets.

| | Model | Average | ImageNet | CIFAR-10 | CIFAR-100 | STL-10 | Food-101 | Caltech-101 | Cars | Aircraft | Flowers | EuroSAT | DTD | Pets | SUN397 |
|---|---|---|---|---|---|---|---|---|---|---|---|---|---|---|---|
| | CLIP | 26.33 | 20.95 | 56.77 | 24.38 | 82.80 | 14.06 | 50.07 | **1.46** | **1.20** | 11.64 | 13.84 | **13.88** | 14.10 | 37.20 |
| CC3M | WPSE | 27.38 | 22.17 | 59.40 | 30.78 | 81.64 | 14.80 | 50.70 | 1.27 | 1.13 | 14.01 | 16.20 | 13.83 | 12.78 | 37.27 |
| | KME-CLIP | **29.31** | **22.66** | **64.31** | **31.32** | **85.23** | **16.17** | **53.67** | 1.29 | 1.04 | **16.54** | **17.68** | 12.39 | **16.26** | **42.49** |
| | CLIP | 43.48 | 36.99 | 75.41 | 42.77 | 92.36 | 45.97 | 71.60 | 17.44 | 2.48 | **25.58** | 31.72 | 19.47 | 55.59 | 47.88 |
| CC12M | WPSE | 43.37 | 36.92 | 74.54 | 42.22 | 92.24 | 43.30 | 71.53 | **18.62** | **3.80** | 22.32 | **32.98** | 19.47 | 54.52 | 51.36 |
| | KME-CLIP | **45.67** | **39.07** | **78.06** | **46.63** | **92.74** | **49.14** | **76.62** | 17.52 | 3.45 | 25.00 | 31.28 | **21.60** | **60.67** | **51.94** |

### 5.3 ABLATION STUDY: REDUCING THE SIZE OF POINT SET

We investigated the impact of the size of point set $m_\mathcal{X}$ in KME-CLIP on retrieval performance using the validation split of CC3M.

To reduce the point set size $m_\mathcal{X}$ in the image encoder, we utilized only a subset of tokens from the beginning of the 197 outputs generated by the ViT architecture. We maintained the original point set size $m_\mathcal{Y}$ for the text encoder without modification. All other training settings followed the configuration described earlier in this section.

We report the top-1 accuracy in Table 3. While the performance of KME-CLIP decreases as the point set size is reduced, it consistently maintains superior performance compared to CLIP even when the point set size is reduced to just 2. See Appendix A.3 for a more detailed comparison including their computational cost. This result also indicates that larger size of point set leads to high performance of KME-CLIP, as is consistent with the theoretical result in Theorem 5.

Table 3: The average of top1 accuracy (%) for text retrieval and image retrieval tasks.

| Size of point set | (CLIP) | 2 | 10 | 50 | 100 | 197 |
|---|---|---|---|---|---|---|
| Top1 accuracy | (23.66) | 23.71 | 23.93 | 24.35 | 24.79 | 25.57 |

## 6 RELATED WORK

We summarize previous work for improving CLIP's capabilities in this section.

**Modification of data.** To address CLIP's requirement for massive training data, Li et al. (2022b) proposed DeCLIP, which incorporates self-supervised learning techniques, probabilistic generation of multiple captions and images through augmentation, and learning from semantically similar captions. Gokhale et al. (2022) introduced specialized datasets to enhance spatial understanding capabilities, an area where CLIP demonstrates notable limitations. Parashar et al. (2024) developed an algorithm to mitigate class imbalances in web-collected training data.

**Modification of loss function.** Yao et al. (2022) proposed FILIP, which maximizes similarity at the token level. Mu et al. (2022) proposed SLIP, which uses in-modal self-supervised learning in addition to the usual contrastive loss. Goel et al. (2022) proposed CyCLIP, which incorporates geometric structure into the alignment of image and text embeddings. Gong et al. (2025) proposed an

algorithm that uses kernel functions to train image embeddings from different pretrained encoders to become more similar to each other. Dong et al. (2023) proposed MaskCLIP, which utilizes distilled representations from masked images.

**Modification of modeling.** Our method falls into this category. Fürst et al. (2022) proposed CLOOB, which employs a Hopfield network for similarity calculation, based on the analogy that humans extract such relationships through memory. Desai et al. (2023) introduced MERU, which embeds representations in hyperbolic space to better capture hierarchical structures. Chou & Alam (2024) proposed EuCLIP, which can capture hierarchical structures without using a geometry of hyperbolic space like MERU. Uesaka et al. (2025) proposed WPSE, which is closely related to our approach; we have discussed the distinctions between our method and WPSE in Section 3.4.

**Other related work.** Ji et al. (2023) provided a theoretical analysis of a linear variant of the contrastive loss in finite dimensions. Finally, while not on contrastive learning, Nishikawa et al. (2025) pointed out that softmax operations in the attention mechanism can be treated within the framework of positive definite kernels, similar to our Proposition 2.

## 7 CONCLUSION

We have proposed a novel approach for refining similarity computation in CLIP, termed KME-CLIP, which leverages kernel mean embeddings in a reproducing kernel Hilbert space. Our theoretical analysis demonstrates that the similarity function in KME-CLIP can approximate PMI—which is theoretically guaranteed to be the optimal similarity—with arbitrary accuracy as the size of point set increases. From an empirical perspective, we demonstrate that KME-CLIP consistently outperforms conventional CLIP in retrieval and zero-shot classification tasks.

## ETHICS STATEMENT

This paper aims to advance machine learning technology through theoretical improvements to CLIP. However, we acknowledge that our method, like other powerful representation learning approaches, may potentially have negative societal impacts if misused. In particular, our method, by increasing the model's representational capacity through infinite-dimensional RKHS embeddings, might inadvertently amplify existing biases if the training data contains biased associations. Therefore, careful consideration must be given to responsible deployment and application of the proposed method.

## REPRODUCIBILITY STATEMENT

We provide comprehensive experimental setting in Section 5 and Appendix A. Additionally, we include our experimental code as supplementary material to ensure full reproducibility of our results.

## LLM USAGE

We utilized large language models (LLMs) to ensure natural academic prose and to assist in debugging and resolving errors in our experimental codebase.

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

## A  ADDITIONAL INFORMATION FOR EXPERIMENTS

### A.1  EXPERIMENTAL SETUP

**Implementation of KME-CLIP.**   To generate multiple point set image embeddings $\{f_i^{\mathcal{X}}(x)\}_{i=1}^{m_{\mathcal{X}}}$, we linearly project the $197 \times 768$ intermediate features of a non-pretrained ViT-B/16 encoder (Dosovitskiy et al., 2021) into a $197 \times 512$ tensor and apply $L^2$-normalization to each embedding. Consequently, we set $m_{\mathcal{X}}$ to 197 and $d$ to 512. To derive the positive weights $\{w_i^{\mathcal{X}}(x)\}_{i=1}^{m_{\mathcal{X}}}$ for image embeddings, we linearly project the $197 \times 768$ intermediate features of ViT into a $197 \times 1$ tensor and pass it through a softplus function, *i.e.*, $\frac{1}{1+e^{-x}}$, to ensure positivity.

For the text embeddings $\{f_i^{\mathcal{Y}}(y)\}_{i=1}^{m_{\mathcal{Y}}}$, we project the $77 \times 512$ output tensor from a non-pretrained Transformer encoder (Vaswani et al., 2017) with width 512, 12 layers, and 8 attention heads into a $77 \times 512$ tensor via linear projection and apply $L^2$-normalization to each embedding. Accordingly, we set $m_{\mathcal{Y}}$ to 77. Regarding the text positive weights $\{w_i^{\mathcal{Y}}(y)\}_{i=1}^{m_{\mathcal{Y}}}$, we linearly project the $77 \times 512$ output tensor of the Transformer into a $77 \times 1$ tensor and pass it through a softplus function to ensure positivity.

We employ the Gaussian kernel for the non-linear kernel function $k$, *i.e.*, $k(u, v) := \exp(-\|u, v\|^2 / 2\sigma^2)$. To avoid numerical instability when $\sigma$ approaches zero, we reparameterize using $\tau = 1/\sigma$ and establish $\tau$ as a learnable parameter in our implementation.

**Implementation of CLIP.**   For the image embedding of CLIP, we projected the 768 dimensional output vector of ViT (it is a projected patch at the position of [CLS] token) into 512 dimensional tensor by a linear projection. For the text embedding of CLIP, we only used the [EOS] embedding of the Transformer and projected it into the 512 dimensional vector by a linear projection.

**Implementation of WPSE.**   As for WPSE, we followed the setting in Uesaka et al. (2025). Moreover, the procedure for obtaining the image and text embeddings $\{f_i^{\mathcal{X}}(x)\}_{i=1}^{m_{\mathcal{X}}}$, $\{f_i^{\mathcal{Y}}(y)\}_{i=1}^{m_{\mathcal{Y}}}$ and weights $\{w_i^{\mathcal{X}}(x)\}_{i=1}^{m_{\mathcal{X}}}$, $\{w_i^{\mathcal{Y}}(y)\}_{i=1}^{m_{\mathcal{Y}}}$ is identical to KME-CLIP except for passing through a softplus function. For the kernel function $k$, we combined Gaussian kernel and linearized kernel using linear combination. To reduce computational cost, we utilized the random Fourier feature for calculating the kernel function.

**Details of hyper-parameter.**   For pretraining, we used the following hyper-parameter presented in Table 4 across CLIP, WPSE, and KME-CLIP on both CC3M and CC12M. For the initialization of the length parameter $\sigma$ in the Gaussian kernel, we set the initial value of $1/\sigma$ in KME-CLIP to 3.78, as this corresponds to WPSE's initial setting of logit scale $\tau = 14.3 \ (= 1/\sigma^2)$.

Table 4: Hyper-parameter for pre-training the encoders

| Config | Value |
| --- | --- |
| Total epochs | 50 |
| Warmup epochs | 2 |
| Warmup start learning rate | $10^{-6}$ |
| Warmup end learning rate | $5.0 \times 10^{-4}$ |
| End learning rate | $10^{-5}$ |
| AdamW $\beta$ | $\beta_1, \beta_2 = (0.9, 0.98)$ |
| AdamW $\epsilon$ | $10^{-8}$ |
| AdamW weight decay | 0.5 |

### A.2  LINEAR CLASSIFICATION

Linear classification is the image classification by logistic regression using features generated by the trained encoder. We evaluated the image encoder's linear classification ability by the same datasets as the zero-shot classification. For CLIP, we use the 768 dimensional output vector of ViT as the feature vector for the logistic regression. For KME-CLIP and WPSE, we extracted the

$197 \times 768$ intermediate features from ViT and calculated the 768 dimensional weighted vector, using the learned weight. Then we use this feature vector as the logistic regression. This procedure is also done in SLIP (Mu et al., 2022).

Following Uesaka et al. (2025), we ran hyperparameter searches over $C \in [10^{-6}, 10^6]$ with a parametric binary search on a validation split of each dataset. We trained a classifier on the combination of training and validation splits and report its performance on the test split.

The result is shown in Table 5. Notably, KME-CLIP demonstrates performance comparable to the other two methods but does not yield significant accuracy improvements. This can be attributed to our use of features extracted before projection into the RKHS, which prevents us from fully leveraging the advantages of our proposed method.

Table 5: Linear classification results. Mean per-class accuracy (%) is reported for Caltech-101, Aircraft, Flowers, and Pets datasets. Top-1 accuracy (%) is used for all other datasets.

|  | Model | Average | ImageNet | CIFAR-10 | CIFAR-100 | STL-10 | Food-101 | Caltech-101 | Cars | Aircraft | Flowers | EuroSAT | DTD | Pets | SUN397 |
|---|---|---|---|---|---|---|---|---|---|---|---|---|---|---|---|
|  | CLIP | 72.29 | 59.59 | 88.75 | 70.33 | 92.94 | 67.02 | 82.34 | 37.45 | 44.20 | 92.02 | **95.88** | 66.01 | 72.88 | 70.32 |
| CC3M | WPSE | 74.29 | 61.34 | 89.51 | 70.89 | **93.63** | **69.80** | **85.22** | 44.35 | 48.51 | 92.57 | 95.10 | **68.03** | **75.22** | **71.64** |
|  | KME-CLIP | **74.44** | **62.02** | **89.73** | **70.94** | 93.10 | 69.76 | 84.62 | **45.69** | **49.28** | **93.16** | 95.80 | 67.82 | 74.64 | 71.13 |
|  | CLIP | 79.45 | 67.79 | 91.23 | 73.96 | 95.56 | 78.79 | 88.50 | 66.55 | 48.77 | 93.46 | 95.74 | **74.41** | 81.58 | 76.50 |
| CC12M | WPSE | **81.17** | **69.71** | **91.98** | 74.78 | **95.69** | **79.81** | **89.93** | **71.45** | **57.60** | **94.93** | 96.02 | 72.66 | **83.62** | **77.05** |
|  | KME-CLIP | 78.31 | 67.76 | 91.92 | **75.24** | 95.41 | 78.28 | 88.13 | 60.22 | 46.54 | 93.52 | **96.16** | 73.72 | 74.89 | 76.20 |

## A.3 COMPUTATIONAL COST

To address computational efficiency concerns, we also investigate the performance of our method when reducing the size of point set $m_\mathcal{X}$ of the image encoder.

For training costs, we measured the number of H100 GPUs required and the time per epoch when training on CC3M. For inference costs, we measured the time required to complete the retrieval task on the validation split of CC3M using two H100 GPUs. We also report the top-1 accuracy for each configuration. The experimental settings follow those described in Section 5.3.

The inference-related results for trained models are presented in Table 6, and the training resource and cost for each configuration is summarized in Table 7. Notably, KME-CLIP generally outperforms CLIP even when operating with a reduced size of point set. When we set the size of point set to 10, KME-CLIP maintains superior performance compared to CLIP while requiring the same number of GPUs and nearly identical training time.

Table 6: Top1 accuracy (%) and inference time required, averaged over text and image retrieval on validation split of CC3M.

| Size of point set | CLIP | 2 | 10 | 50 | 100 | 197 |
|---|---|---|---|---|---|---|
| Accuracy (%) | 23.66 | 23.71 | 23.93 | 24.35 | 24.79 | 25.57 |
| Inference time with two GPUs (sec) | 126 | 138 | 140 | 159 | 186 | 257 |

Table 7: Number of H100 GPUs and time required for training by KME-CLIP and CLIP on CC3M.

| Size of point set | CLIP | 2 | 10 | 50 | 100 | 197 |
|---|---|---|---|---|---|---|
| Training time (sec/epoch) | 1604 | 1413 | 1439 | 925 | 722 | 945 |
| Number of GPUs for training | 2 | 2 | 2 | 4 | 8 | 8 |

## B    PROOF OF AUXILIARY RESULTS

### B.1    PROOF OF PROPOSITION 2

*Proof.* Under the given setting, the similarity of KME-CLIP is

$$S(x,y) = \log \sum_{i,j} w_i^{\mathcal{X}}(x) w_j^{\mathcal{Y}}(y) k(f_i^{\mathcal{X}}(x), f_j^{\mathcal{Y}}(y)) \tag{5}$$

$$= \log(\exp(-\left\| f_1^{\mathcal{X}}(x) - f_1^{\mathcal{Y}}(y) \right\|^2 / 2\sigma^2)) \tag{6}$$

$$= \frac{-\left\| f_1^{\mathcal{X}}(x) \right\|^2 - \left\| f_1^{\mathcal{Y}}(y) \right\|^2 + 2 f_1^{\mathcal{X}}(x)^\top f_1^{\mathcal{Y}}(y)}{2\sigma^2} \tag{7}$$

$$= -\frac{1}{\sigma^2} + f_1^{\mathcal{X}}(x)^\top f_1^{\mathcal{Y}}(y)/\sigma^2, \tag{8}$$

where the last equality derives from the $L^2$ normalization when calculating the embedding. Since we set $\tau = \sigma^2$, taking $g^{\mathcal{X}} = f_1^{\mathcal{X}}$ and $g^{\mathcal{Y}} = f_1^{\mathcal{Y}}$ concludes the desired. □

## C    PROOF OF THEOREMS IN SECTION 4

### C.1    PROOF OF THEOREM 3

We start from the following lemma.

**Lemma 8.** *Let $\mathcal{S}$ be a measure space associated with a finite measure $\mu$. Let $\epsilon \geq \delta > 0$ and $f, g : \mathcal{S} \to \mathbb{R}_+$ satisfy the following:*

- *$\epsilon \leq 1/e^2$,*

- *$f$ is a density function with respect to $\mu$,*

- *$g$ satisfies $|f(s) - g(s)| \leq \epsilon$ and $g(s) \geq \delta$ for all $s \in \mathcal{S}$.*

*Then, we have*

$$\left| \int_{\mathcal{S}} f(s)(\log f(s) - \log g(s)) \, d\mu(s) \right| \leq 2\sqrt{\epsilon} + \mu(\mathcal{S})\sqrt{\epsilon} \log \left( \frac{1}{\epsilon} + \frac{1}{\delta} \right).$$

*Proof.* Let $A = \{s \in \mathcal{S} \mid f(s) \leq \sqrt{\epsilon}\}$. We decompose the integral into three terms:

$$\int_{\mathcal{S}} f(s)(\log f(s) - \log g(s)) \, d\mu(s)$$

$$= \int_A f(s) \log f(s) \, d\mu(s) - \int_A f(s) \log g(s) \, d\mu(s) + \int_{\mathcal{S} \setminus A} f(s)(\log f(s) - \log g(s)) \, d\mu(s). \tag{9}$$

For the first term, since $\sqrt{\epsilon} < 1/e$ and $|x \log x|$ is monotonically increasing over $(0, 1/e)$, we have

$$\left| \int_A f(s) \log f(s) \, d\mu(s) \right| \leq \int_A |\sqrt{\epsilon} \log \sqrt{\epsilon}| \, d\mu(s) \leq \frac{\mu(\mathcal{S})}{2} \sqrt{\epsilon} \log \frac{1}{\epsilon}. \tag{10}$$

For the second term, for each $s \in A$, we have $\delta \leq g(s) \leq f(s) + \epsilon \leq \sqrt{\epsilon} + \epsilon \leq 1/e + 1/e^2 < 1$. Thus, we obtain

$$\left| \int_A f(s) \log g(s) \, d\mu(s) \right| \leq \int_A f(s) \log \frac{1}{\delta} \, d\mu(s) \leq \mu(\mathcal{S})\sqrt{\epsilon} \log \frac{1}{\delta}. \tag{11}$$

For the final term of (9), we need to bound $|\log f(s) - \log g(s)|$. For a positive constant $a > 0$, $\log(a + x)$ is $(1/a)$-Lipschitz continuous for $x \geq 0$. By using this fact, since $f(s) \geq \sqrt{\epsilon}$ and $g(s) \geq \sqrt{\epsilon} - \epsilon$ for $s \in \mathcal{S} \setminus A$, we have

$$|\log f(s) - \log g(s)| \leq \frac{|f(s) - g(s)|}{\sqrt{\epsilon} - \epsilon} \leq \frac{\epsilon}{\sqrt{\epsilon} - \epsilon} = \frac{\sqrt{\epsilon}}{1 - \sqrt{\epsilon}} \leq \frac{\sqrt{\epsilon}}{1 - 1/e} < 2\sqrt{\epsilon}.$$

Therefore, we have

$$\left| \int_{\mathcal{S} \setminus A} f(s)(\log f(s) - \log g(s)) \, \mathrm{d}\mu(s) \right| \leq \int_{\mathcal{S} \setminus A} f(s) 2\sqrt{\epsilon} \, \mathrm{d}\mu(s) \leq 2\sqrt{\epsilon}. \tag{12}$$

By applying (10)–(12) to (9) and rearranging some constants, we obtain the desired estimate. □

Let us now prove the theorem.

*Proof of Theorem 3.* From Theorem 1, $S = \log \frac{p(x,y)}{p(x)p(y)}$ attains the minimum of $L_S$. Let $f(x,y) = \frac{p(x,y)}{p(x)p(y)}$. We fist estimate the error for first term of $L_S$:

$$\left| \mathbb{E}_{p(x,y)} \left[ \log \frac{f(x,y)}{\mathbb{E}_{p(x')}[f(x',y)]} \right] - \mathbb{E}_{p(x,y)} \left[ \log \frac{h(x,y)}{\mathbb{E}_{p(x')}[h(x',y)]} \right] \right|$$
$$\leq \left| \mathbb{E}_{p(x,y)} [\log f(x,y) - \log h(x,y)] \right| + \left| \mathbb{E}_{p(y)} [\log(\mathbb{E}_{p(x)}[f(x,y)]) - \log(\mathbb{E}_{p(x)}[h(x,y)])] \right|. \tag{13}$$

By rewriting the first term of (13) by integral, we obtain the following:

$$\left| \mathbb{E}_{p(x,y)} [\log f(x,y) - \log h(x,y)] \right|$$
$$= \left| \iint_{\mathcal{X} \times \mathcal{Y}} p(x,y) \log f(x,y) \, \mathrm{d}\nu_X(x) \, \mathrm{d}\nu_Y(y) - \iint_{\mathcal{X} \times \mathcal{Y}} p(x,y) \log h(x,y) \, \mathrm{d}\nu_X(x) \, \mathrm{d}\nu_Y(y) \right|$$
$$= \left| \iint_{\mathcal{X} \times \mathcal{Y}} f(x,y) \log f(x,y) \, \mathrm{d}\tilde{\nu}_X(x) \, \mathrm{d}\tilde{\nu}_Y(y) - \iint_{\mathcal{X} \times \mathcal{Y}} f(x,y) \log h(x,y) \, \mathrm{d}\tilde{\nu}_X(x) \, \mathrm{d}\tilde{\nu}_Y(y) \right|,$$

where $\mathrm{d}\tilde{\nu}_X(x) = p(x) \, \mathrm{d}\nu_X(x)$ and $\mathrm{d}\tilde{\nu}_Y(y) = p(y) \, \mathrm{d}\nu_X(y)$. Not also that we have

$$\tilde{\nu}_X(\mathcal{X}) = \int_{\mathcal{X}} p(x) \, \mathrm{d}\nu_X(x) = 1, \qquad \tilde{\nu}_Y(\mathcal{Y}) = \int_{\mathcal{X}} p(y) \, \mathrm{d}\nu_Y(y) = 1,$$

since $p(x)$ and $p(y)$ are density functions, and $f(x,y)$ is a density function with respect to $\tilde{\nu}_X \otimes \tilde{\nu}_Y$, because

$$\iint_{\mathcal{X} \times \mathcal{Y}} f(x,y) \, \mathrm{d}\tilde{\nu}_X(x) \, \mathrm{d}\tilde{\nu}_Y(y) = \iint_{\mathcal{X} \times \mathcal{Y}} p(x,y) \, \mathrm{d}\nu_X(x) \, \mathrm{d}\nu_Y(y) = 1.$$

Thus, by applying Lemma 8 for the product measure $\tilde{\nu}_X \otimes \tilde{\nu}_Y$, we have

$$\left| \mathbb{E}_{p(x,y)} [\log f(x,y) - \log h(x,y)] \right| \leq 2\sqrt{\epsilon} + \tilde{\nu}_X(\mathcal{X})\tilde{\nu}_Y(\mathcal{Y})\sqrt{\epsilon} \log \left( \frac{1}{\epsilon} + \frac{1}{\delta} \right)$$
$$= 2\sqrt{\epsilon} + \sqrt{\epsilon} \log \left( \frac{1}{\epsilon} + \frac{1}{\delta} \right). \tag{14}$$

Next, let us estimate the second term of (13). Let $F_Y(y) := \mathbb{E}_{p(x)}[f(x,y)]$ and $H_Y(y) := \mathbb{E}_{p(x)}[h(x,y)]$. Since they are expectations, we can see that the assumed inequality constraints is inherited to $H_Y$ and $F_Y$, *i.e.*, $|H_Y(y) - F_Y(y)| \leq \epsilon$. Note also that $F_Y(y) = 1$ holds almost everywhere with respect to $\tilde{\nu}_Y$. Indeed, for any measurable set $A \subset \mathcal{Y}$, we have

$$\int_A F_Y(y) \, \mathrm{d}\tilde{\nu}_Y(y) = \int_A p(y) \int_{\mathcal{X}} p(x) f(x,y) \, \mathrm{d}\nu_X(x) \, \mathrm{d}\nu_Y(y)$$
$$= \iint_{\mathcal{X} \times A} p(x,y) \, \mathrm{d}(\nu_X \otimes \nu_Y)(x,y).$$
$$= \mathbb{P}(Y \in A) = \int_A p(y) \, \mathrm{d}\nu_Y(y).$$

Thus, $H_Y(y) \in [1 - \epsilon, 1 + \epsilon]$ almost everywhere with respect to $\tilde{\nu}$. In particular, we have an integral estimate $\int_{\mathcal{Y}} |\log H_Y(y)| \, \mathrm{d}\tilde{\mu}_Y(y) \leq \log \frac{1}{1-\epsilon}$. By using these, we can evaluate the second term of (13) as follows:

$$\left| \mathbb{E}_{p(y)} [\log(\mathbb{E}_{p(x)}[f(x,y)]) - \log(\mathbb{E}_{p(x)}[h(x,y)])] \right|$$

$$= \left| \mathbb{E}_{p(y)}[\log(F_Y(y)) - \log(H_Y(y))] \right|$$

$$= \left| \int_{\mathcal{Y}} p(y) \log F_Y(y) \, \mathrm{d}\nu_Y(y) - \int_{\mathcal{Y}} p(y) \log H_Y(y) \, \mathrm{d}\nu_Y(y) \right|$$

$$= \left| \int_{\mathcal{Y}} \log H_Y(y) \, \mathrm{d}\tilde{\nu}_Y(y) \right|$$

$$\leq \int_{\mathcal{Y}} |\log H_Y(y)| \, \mathrm{d}\tilde{\mu}_Y(y) \leq \log \frac{1}{1-\epsilon}.$$

Since $\frac{\mathrm{d}}{\mathrm{d}x} \log \frac{1}{1-x} = \frac{1}{1-x}$, the function $\log \frac{1}{1-x}$ is $\frac{1}{1-1/e^2}$-Lipschitz continuous for $0 \leq x \leq 1/e^2$. Thus, we have

$$\left| \mathbb{E}_{p(y)}[\log(\mathbb{E}_{p(x)}[f(x,y)])] - \log(\mathbb{E}_{p(x)}[h(x,y)])] \right|$$

$$\leq \log \frac{1}{1-\epsilon} \leq \frac{1}{1-1/e^2}\epsilon = \frac{\sqrt{\epsilon}}{1-1/e^2}\sqrt{\epsilon} \leq \frac{1/e}{1-1/e^2}\sqrt{\epsilon} = \frac{1}{e-1/e}\sqrt{\epsilon} \leq \sqrt{\epsilon}, \quad (15)$$

where we have used the assumption $\epsilon \leq 1/e^2$ multiple times. By applying (14) and (15) to (13), we can bound the error of the first term (without $1/2$) of $L_S$ by the right-hand side of (4). By conducting the same argument on the second term and taking average, we obtain the desired estimate for $L_S$. $\qquad \square$

## C.2 PROOF OF THEOREM 4

*Proof.* First, since $\int_{\mathcal{Z}} p(x|z)p(z) \, \mathrm{d}\rho(z) = 1$, there exists a $z$ such that $p(x|z)p(z) \leq 1$. Thus, from the Lipschitz continuity and the compactness of $\mathcal{Z}$, $p(x|z)/p(x)$ is bounded by a universal constant. We can show the same for $p(y|z)p(z)$, so there is a universal constant $C'$ such that

$$\frac{p(x|z)}{p(x)}, \frac{p(y|z)}{p(y)} \leq C'.$$

Since $\mathcal{Z}$ is compact, for any $q > 0$, there is a finite covering $\mathcal{Z} = B_1 \cup \cdots \cup B_m$ such that each $B_i$ is an open ball of radius $q/2$ in $\mathcal{Z}$. Let $A_i := B_i \setminus \bigcup_{j<i} B_j$. Then, each $A_i$ is Borel measurable and $\mathcal{Z} = A_1 \cup \cdots \cup A_m$ becomes a disjoint union. We assume each $A_i$ is nonempty; otherwise we simply omit such $A_i$. Since $p(x|z)/p(x)$ and $p(y|z)/p(y)$ are uniformly $C_L$-Lipschitz continuous with respect to $z$, there exists $a^x_{A_i}, a^y_{A_i} \in (0, C']$ satisfying

$$\left| \frac{p(x|z)}{p(x)} - \sum_{i=1}^m a^x_{A_i} \mathbb{1}[z \in A_i] \right| \leq C_L q, \qquad \left| \frac{p(y|z)}{p(y)} - \sum_{i=1}^m a^y_{A_i} \mathbb{1}[z \in A_i] \right| \leq C_L q, \quad (16)$$

for all $x, y, z$. Indeed, for a $C_L$-Lipschitz function $f$ and $z, z' \in A_i$, we have

$$|f(z) - f(z')| \leq C_L d(z, z') \leq C_L \cdot 2 \cdot q/2 = C_L q,$$

where $d$ is the metric function over $\mathcal{Z}$. By taking any $z_i \in A_i$ for each $i$, we thus have

$$\left| f(z) - \sum_{i=1}^m f(z_i)\mathbb{1}[z \in A_i] \right| \leq C_L q,$$

because $A_i$ is disjoint to each other. This justifies (16).

Hence, we have

$$\left| \int_{\mathcal{Z}} \frac{p(x|z)p(y|z)}{p(x)p(y)} \, \mathrm{d}\rho(z) - \sum_{i=1}^m (a^x_{A_i} a^y_{A_i})\rho(A_i) \right|$$

$$\leq \left| \int_{\mathcal{Z}} \left( C_L q + \sum_{i=1}^m a^x_{A_i} \mathbb{1}[z \in A_i] \right) \left( C_L q + \sum_{i=1}^m a^y_{A_i} \mathbb{1}[z \in A_i] \right) \, \mathrm{d}\rho(z) - \sum_{i=1}^m (a^x_{A_i} a^y_{A_i})\rho(A_i) \right|$$

$$= \left| (C_L q)^2 + C_L q \sum_{i=1}^m (a^x_{A_i} + a^y_{A_i})\rho(A_i) + \sum_{i=1}^m (a^x_{A_i} a^y_{A_i})\rho(A_i) - \sum_{i=1}^m (a^x_{A_i} a^y_{A_i})\rho(A_i) \right|$$

$$= (C_L q)^2 + C_L q \sum_{i=1}^{m} (a_{A_i}^x + a_{A_i}^y) \rho(A_i).$$

$$\leq (C_L q)^2 + 2 C_L C' q \sum_{i=1}^{m} \rho(A_i) = (C_L q)^2 + 2 C_L C' q \tag{17}$$

Let us construct $\mu$ and embedding functions now. We take $m$ distinct points $u_i$ in $\mathbb{R}^d$. Even if we are constrained in the unit sphere ($d \geq 2$), we can take $u_i$'s so that, for $i \neq j$ we have $\|u_i - u_j\| \geq 2/m$ by manipulating one of the coordinates. Let $\mu$ be the uniform measure over $\{u_1, \ldots u_m\}$, *i.e.*, $\mu = \frac{1}{m} \sum_{i=1}^{m} \delta_{u_i}$.

If we set $g_X^x(u_i) = a_{A_i}^x \sqrt{m \rho(A_i)}$ and $g_Y^y(u_i) = a_{A_i}^y \sqrt{m \rho(A_i)}$ for $i = 1, \ldots, m$, they can be regarded as $L^2$ functions with respect to the measure $\mu$. Then, we have

$$\left| \left\langle \int_{z \in \mathcal{Z}} g_X^x(z) k(u, \cdot) \, \mathrm{d}\mu(u), \int_{z \in \mathcal{Z}} g_Y^y(u) k(u, \cdot) \, \mathrm{d}\mu(u) \right\rangle_{\mathcal{H}} - \sum_{i=1}^{m} (a_{A_i}^x a_{A_i}^y) \rho(A_i) \right|$$

$$= \left| \left\langle \frac{1}{m} \sum_{i=1}^{m} a_{A_i}^x \sqrt{m \rho(A_i)} k(u_i, \cdot), \frac{1}{m} \sum_{i=1}^{m} a_{A_i}^y \sqrt{m \rho(A_i)} k(u_i, \cdot) \right\rangle_{\mathcal{H}} - \sum_{i=1}^{m} (a_{A_i}^x a_{A_i}^y) \rho(A_i) \right|$$

$$= \left| \frac{1}{m} \sum_{i=1}^{m} (a_{A_i}^x a_{A_i}^y) \rho(A_i) k(z_i, z_i) + \frac{1}{m} \sum_{i \neq j} (a_{A_i}^x a_{A_j}^y) \sqrt{\rho(A_i)} \sqrt{\rho(A_j)} k(z_i, z_j) - \sum_{i=1}^{m} (a_{A_i}^x a_{A_i}^y) \rho(A_i) \right|$$

$$\leq \frac{1}{m} \sum_{i \neq j} (a_{A_i}^x a_{A_j}^y) \sqrt{\rho(A_i)} \sqrt{\rho(A_j)} \exp(-(2/m)^2 / (2\sigma^2))$$

$$\leq \frac{(C')^2}{m} \sum_{i,j=1}^{m} \sqrt{\rho(A_i)} \sqrt{\rho(A_j)} \exp(-2/(m\sigma)^2)$$

$$\leq \frac{(C')^2}{m} \sum_{i=1}^{m} \rho(A_i) \sum_{j=1}^{m} \rho(A_j) \exp(-2/(m\sigma)^2)$$

$$= \frac{(C')^2}{m} \exp(-2/(m\sigma)^2), \tag{18}$$

where we have used the Cauchy–Schwarz in the last inequality.

Combining (17) and (18), we have

$$\left| \left\langle \int_{z \in \mathcal{Z}} g_X^x(z) k(z, \cdot) \, \mathrm{d}\mu(z), \int_{z \in \mathcal{Z}} g_Y^y(z) k(z, \cdot) \, \mathrm{d}\mu(z) \right\rangle_{\mathcal{H}} - \frac{p(x,y)}{p(x)p(y)} \right|$$

$$\leq (C_L q)^2 + 2 C_L C' q + \frac{(C')^2}{m} \exp(-2/(m\sigma)^2). \tag{19}$$

If we take $q < \frac{\epsilon}{4 \max\{C_L, C_L C'\}}$ and then $\sigma$ to be sufficiently small, the right-hand side of (19) becomes smaller than $\epsilon$, which concludes the proof.

$\square$

### C.3 PROOF OF THEOREM 5

*Proof.* We can show the result from straight-forward calculation about the norm in RKHS. Let $f = \int_{\mathcal{Z}} k(u, \cdot) g(u) \, \mathrm{d}\mu(u)$. Then, for any $u_1, \ldots, u_{m_{\mathcal{X}}} \in \mathbb{R}^d$, we have

$$\left\| f - \sum_{i=1}^{m_{\mathcal{X}}} \frac{1}{m_{\mathcal{X}}} k(u_i, \cdot) g(u_i) \right\|_{\mathcal{H}}^2$$

$$= \|f\|_{\mathcal{H}}^2 + \left\| \sum_{i=1}^{m_{\mathcal{X}}} \frac{1}{m_{\mathcal{X}}} k(u_i, \cdot) g(u_i) \right\|_{\mathcal{H}}^2 - 2 \left\langle f, \sum_{i=1}^{m_{\mathcal{X}}} \frac{1}{m_{\mathcal{X}}} k(u_i, \cdot) g(u_i) \right\rangle_{\mathcal{H}}$$

$$= \iint k(u,u')g(u)g(u')\,\mathrm{d}\mu(u)\,\mathrm{d}\mu(u') + \frac{1}{m_{\mathcal{X}}^2}\sum_{i=1}^{m_{\mathcal{X}}}\sum_{j=1}^{m_{\mathcal{X}}} k(u_i,u_j)g(u_i)g(u_j) - 2\sum_{i=1}^{m_{\mathcal{X}}}\frac{1}{m_{\mathcal{X}}}f(u_i)g(u_i)$$

$$= \left( \iint k(u,u')g(u)g(u')\,\mathrm{d}\mu(u)\,\mathrm{d}\mu(u') - \frac{1}{m_{\mathcal{X}}}\sum_{i=1}^{m_{\mathcal{X}}}\int k(u_i,u)g(u)g(u_i)\,\mathrm{d}\mu(u) \right)$$

$$+ \left( \frac{1}{m_{\mathcal{X}}^2}\sum_{i=1}^{m_{\mathcal{X}}}\sum_{j=1}^{m_{\mathcal{X}}} k(u_i,u_j)g(u_i)g(u_j) - \frac{1}{m_{\mathcal{X}}}\sum_{i=1}^{m_{\mathcal{X}}}\int k(u_i,u)g(u)g(u_i)\,\mathrm{d}\mu(u) \right). \tag{20}$$

It is enough to show that the expected value of the right-hand side of (20) have the upper bound $K\,\|g\|_{L^2(\mu)^2}\,/m_{\mathcal{X}}$ when we sample $u_1,...,u_{m_{\mathcal{X}}} \sim \mu$.

As for the first term in the right-hand side of (20), we have

$$\mathbb{E}\left[ \iint k(u,u')g(u)g(u')\,\mathrm{d}\mu(u)\,\mathrm{d}\mu(u') - \frac{1}{m_{\mathcal{X}}}\sum_{i=1}^{m_{\mathcal{X}}}\int k(u_i,u)g(u)g(u_i)\,\mathrm{d}\mu(u) \right] = 0.$$

Also, as for the second term in the right-hand side of (20), we have

$$\mathbb{E}\left[ \frac{1}{m_{\mathcal{X}}^2}\sum_{i,j} k(u_i,u_j)g(u_i)g(u_j) - \frac{1}{m_{\mathcal{X}}}\sum_{i=1}^{m_{\mathcal{X}}}\int k(u_i,u)g(u)g(u_i)\,\mathrm{d}\mu(u) \right]$$

$$= \frac{1}{m_{\mathcal{X}}^2}\mathbb{E}\left[ \sum_{i=1}^{m_{\mathcal{X}}} k(u_i,u_i)g(u_i)g(u_i) \right]$$

$$+ \frac{1}{m_{\mathcal{X}}^2}\mathbb{E}\left[ \sum_{i \neq j} k(u_i,u_j)g(u_i)g(u_j) \right] - \iint_{\mathcal{Z}} k(u',u)g(u)g(u')\,\mathrm{d}\mu(u)\,\mathrm{d}\mu(u')$$

$$= \frac{1}{m_{\mathcal{X}}^2}\mathbb{E}\left[ \sum_{i=1}^{m_{\mathcal{X}}} k(u_i,u_i)g(u_i)g(u_i) \right] - \frac{1}{m_{\mathcal{X}}}\iint k(u',u)g(u)g(u')\,\mathrm{d}\mu(u)\,\mathrm{d}\mu(u').$$

In the last equality, we used $\mathbb{E}[k(u_i,u_j)g(u_i)g(u_j)] = \iint_{\mathcal{Z}\times\mathcal{Z}} k(z,z')g(z)g(z')\,\mathrm{d}\mu(z)\,\mathrm{d}\mu(z')$ for $i \neq j$, which is positive from the positive-definiteness of $k$. Thus, the final term is bounded by

$$\frac{1}{m_{\mathcal{X}}^2}\mathbb{E}\left[ \sum_{i=1}^{m_{\mathcal{X}}} k(u_i,u_i)g(u_i)g(u_i) \right] = \frac{1}{m_{\mathcal{X}}}\int k(u,u)g(u)^2\,\mathrm{d}\mu(u) \leq \frac{1}{m_{\mathcal{X}}}K\,\|g\|_{L^2}^2\,,$$

which is the desired bound. $\qquad\square$

### C.4 Proof of Theorem 6

*Proof.* In this proof, let us $(i,j)$ as $ij$ for notational simplicity. We only have to show that if condition 1. does not hold, i.e., if low modality gap condition holds, then condition 2. holds. We give a proof by contradiction, with the assumption that, for each $ij$ and $k\ell$, we have

$$|\alpha \exp\left(S(ij,k\ell)\right) - \exp(\mathrm{PMI}(ij,k\ell))| < \frac{N}{4}. \tag{21}$$

From Assumption 4, we have $p_X(ij) = p_Y(ij) = \frac{2}{N(N+1)}$ for all $i,j \in \mathcal{Z}$. Thus, we have

$$\frac{p_{X|Z}(ij|\ell)}{p_X(ij)} = \frac{N(N+1)}{2}\frac{\delta^{i\ell} + \delta^{j\ell}}{N+1} = \frac{N}{2}(\delta^{i\ell} + \delta^{j\ell}). \tag{22}$$

The same holds for $p_Y$. Thus, for $i \neq j$, we have identities such as

$$\exp(\mathrm{PMI}(ii,jj)) = 0, \quad \exp(\mathrm{PMI}(ii,ij)) = \frac{1}{N}\cdot\frac{2N}{2}\cdot\frac{N}{2} = \frac{N}{2}, \quad \exp(\mathrm{PMI}(ii,ii)) = N. \tag{23}$$

These also hold for $\text{PMI}^{\mathcal{X}}$. In particular, given $ii \in \mathcal{X}$ we have $S(ii, ii) \geq S(ii, j\ell)$ for any $j\ell$ because of (21). Indeed, $\alpha \exp(S(ii, ii)) > 3N/4$ should be satisfied from (21) and (23), but for any other $j\ell$ we have $\alpha \exp(S(ii, j\ell)) < 3N/4$ from the same reasoning. Considering that $\exp(\text{PMI}(ii, jj)) = 0 < \delta$ for any $\delta > 0$, we have the following from a low modality gap condition:

$$S^{\mathcal{X}}(ii, jj) \leq S(ii, ii). \tag{24}$$

When $u, v \in \mathbb{R}^d$ are on the unit sphere, we have

$$\alpha \exp(u^\top v / \tau) = \alpha \exp\left( \frac{-\|u - v\|^2 + \|u\|^2 + \|v\|^2}{2\tau} \right) = \alpha \exp(1/\tau) \exp\left( -\frac{\|u - v\|^2}{2\tau} \right).$$

Thus, by letting $\alpha_0 := \alpha \exp(1/\tau)$, we have

$$\alpha \exp(S^{\mathcal{X}}(x, x')) = \alpha_0 \exp\left( -\frac{\left\| g^{\mathcal{X}}(x) - g^{\mathcal{X}}(x') \right\|^2}{2\tau} \right),$$

$$\alpha \exp(S(x, y)) = \alpha_0 \exp\left( -\frac{\left\| g^{\mathcal{X}}(x) - g^{\mathcal{Y}}(y) \right\|^2}{2\tau} \right). \tag{25}$$

From these, we can translate (24) to $\|g^{\mathcal{X}}(ii) - g^{\mathcal{X}}(jj)\| \geq \|g^{\mathcal{X}}(ii) - g^{\mathcal{Y}}(ii)\|$. From the triangle inequality, we also have

$$\|g^{\mathcal{X}}(ii) - g^{\mathcal{X}}(jj)\| \geq \|g^{\mathcal{X}}(jj) - g^{\mathcal{Y}}(ii)\| - \|g^{\mathcal{X}}(ii) - g^{\mathcal{Y}}(ii)\|.$$

By taking the average of these two lower bounds, we have

$$\|g^{\mathcal{X}}(ii) - g^{\mathcal{X}}(jj)\| \geq \frac{\|g^{\mathcal{X}}(ii) - g^{\mathcal{Y}}(ii)\| + \|g^{\mathcal{X}}(jj) - g^{\mathcal{Y}}(ii)\| - \|g^{\mathcal{X}}(ii) - g^{\mathcal{Y}}(ii)\|}{2}$$

$$= \frac{\|g^{\mathcal{X}}(jj) - g^{\mathcal{Y}}(ii)\|}{2} = \frac{1}{2} \sqrt{2\tau \log \frac{\alpha_0}{\alpha \exp(S(jj, ii))}}$$

$$\geq \frac{1}{2} \sqrt{2\tau \log \frac{\alpha_0}{\exp(\text{PMI}(jj, ii)) + N/4}} = \frac{1}{2} \sqrt{2\tau \log \frac{\alpha_0}{N/4}}, \tag{26}$$

where we have used (21), (23), and (25).

We can also bound the opposite direction as

$$\|g^{\mathcal{X}}(ii) - g^{\mathcal{X}}(jj)\| \leq \|g^{\mathcal{X}}(ii) - g^{\mathcal{Y}}(ij)\| + \|g^{\mathcal{X}}(jj) - g^{\mathcal{Y}}(ij)\|$$

$$= \sqrt{2\tau \log \frac{\alpha_0}{\alpha \exp(S(ii, ij))}} + \sqrt{2\tau \log \frac{\alpha_0}{\alpha \exp(S(jj, ij))}}$$

$$\leq 2 \sqrt{2\tau \log \frac{\alpha_0}{\exp(\text{PMI}(ii, ij)) - N/4}} = 2 \sqrt{2\tau \log \frac{\alpha_0}{N/4}}. \tag{27}$$

Thus, by letting $R := \sqrt{2\tau \log \frac{\alpha_0}{N/4}}$, we have $R/2 \leq \|g^{\mathcal{X}}(ii) - g^{\mathcal{X}}(jj)\| \leq 2R$ from (26) and (27).

Therefore, if we let $B_i$ be an open ball of radius $R/4$ centered at $g^{\mathcal{X}}(ii)$, $B_1, \ldots, B_N$ are disjoint to each other and included in an ball of radius $(2 + \frac{1}{4})R$, centered at a certain $g^{\mathcal{X}}(\ell\ell)$. By summing up the Lebesgue measure of $N$ balls, we have $N(1/4)^d S \leq (9/4)^d S$, where $S$ is the measure of a unit ball in $\mathbb{R}^d$. Therefore, $N$ needs to satisfy $N \leq 9^d$. $\qquad \square$

### C.5 PROOF OF THEOREM 7

*Proof.* We follow the notation of the proof of Theorem 6. Recall (22) stating the following:

$$\frac{p_{X|Z}(ij|\ell)}{p_X(ij)} = \frac{p_{Y|Z}(ij|\ell)}{p_Y(ij)} = \frac{N}{2}(\delta^{i\ell} + \delta^{j\ell}).$$

From this, we can write the exponential PMI as follows:

$$\exp(\text{PMI}(ij, st)) = \frac{1}{N} \sum_{\ell=1}^{N} \frac{p_{X|Z}(ij|\ell)}{p_X(ij)} \frac{p_{X|Z}(ij|\ell)}{p_X(ij)}$$

$$= \frac{N}{4} \sum_{\ell=1}^{N} (\delta^{i\ell} + \delta^{j\ell})(\delta^{s\ell} + \delta^{t\ell}) = \frac{N}{4}(\delta^{is} + \delta^{it} + \delta^{js} + \delta^{jt}). \qquad (28)$$

Following the idea of Appendix C.2, let us take $N$ distinct points $u_1, \ldots, u_N$ on the unit sphere of $\mathbb{R}^d$, such that $\|u_i - u_j\| \geq 1/N$ for each $i \neq j$. We realize the embedding by

$$h_\theta^{\mathcal{X}}(ij) = h_\theta^{\mathcal{Y}}(ij) = \frac{\sqrt{N}}{2}(k(u_i, \cdot) + k(u_j, \cdot)).$$

The similarities induced by these RKHS embeddings satisfies a low modality gap condition in Definition 2, since $S^{\mathcal{X}}(x, x') = S(x, x')$ holds for every $x, x' \in \mathcal{X}$. Then, we have

$$\langle h_\theta^{\mathcal{X}}(ij), h_\theta^{\mathcal{Y}}(st) \rangle_{\mathcal{H}} = \frac{N}{4}(k(u_i, u_s) + k(u_i, u_t) + k(u_j, u_s) + k(u_j, u_t)). \qquad (29)$$

By the term-by-term comparison of (28) and (29), proving $|k(u_i, u_\ell) - \delta^{i\ell}| < \epsilon/N$ for each $i, \ell$ is sufficient for achieving the desired error bound. Because $k$ is a Gaussian kernel, we have $k(u_i, u_i) = 1$. Therefore, it suffices to prove that we can let $k(u_i, u_\ell) < \epsilon/N$ when $i \neq \ell$. Since we have

$$k(u_i, u_\ell) = \exp\left(-\frac{\|u_i - u_\ell\|^2}{2\sigma^2}\right) \leq \exp\left(-\frac{1}{2N^2\sigma^2}\right)$$

from the construction, just letting $\sigma < (2N^2 \log(N/\epsilon))^{-1/2}$ gives the desired conclusion. $\qquad \square$

