# OpenReview forum: "Theoretical refinement of CLIP by utilizing linear structure of optimal similarity"
_ICLR.cc/2026/Conference — Submitted to ICLR 2026_

### Official Review · Reviewer_4ttG · 2025-10-20

**Soundness:** 2
**Presentation:** 3
**Contribution:** 2
**Rating:** 6
**Confidence:** 3

**Summary:**

This paper identifies that under certain condition - pointwise mutual information (PMI) possesses a linear structure in an inner product of L2 norm which original CLIP fails to exploit. The authors proposes a KME-CLIP that projects embedding into a Kernel Hilbert space where they can exploit those PMI information.

**Strengths:**

1. Finding that under certain condition - pointwise mutual information (PMI) possesses a linear structure in an inner product of L2 norm which original CLIP work fails to exploit and thereby proposing a solution - is a novel contribution
2. The paper is concise and well written

**Weaknesses:**

1. I think a bit more insight on the related work is helpful. For example, this work does not refer a related work https://arxiv.org/abs/2409.13079 and how this might be different
2. The results in Table 1 shows the authors approach is better in most benchmarks compared to CLIP or WPSE. But it is hard to understand the actual impact without some quantitative example.
3. The paper has novel contribution but not clear how impactful the contribution is. What would it take application builders to reject CLIP and start using KME-CLIP?

**Questions:**

Please see the weakness

---

> ### Author Response · Authors · 2025-11-21
>
> We are grateful for your helpful and insightful review comments. We also appreciate your positive feedback on our identification of the structural limitations of CLIP in this paper.
>
> ---
>
> > I think a bit more insight on the related work is helpful. For example, this work does not refer a related work https://arxiv.org/abs/2409.13079 and how this might be different
>
> Thank you for pointing out this related work that we had not previously included. We have now added the paper you mentioned to the related work section.
> The key difference between their work and ours lies in the underlying motivation: while their study is driven by geometric insights into effective similarity computation, our approach is motivated by leveraging the linear structure of PMI within the similarity computation.
>
> ---
>
> > The results in Table 1 shows the authors approach is better in most benchmarks compared to CLIP or WPSE. But it is hard to understand the actual impact without some quantitative example.
>
> > The paper has novel contribution but not clear how impactful the contribution is. What would it take application builders to reject CLIP and start using KME-CLIP?
>
> Theoretically, Theorems 6 and 7 show that CLIP cannot simultaneously satisfy both the approximation of PMI and a low modality gap, whereas KME-CLIP can achieve both properties in a toy setting. We believe that our proof that CLIP actually cannot work well in a simple toy setting like Assumption 4 (which has an intuitive background as written after the assumption) significantly contributes to the field, together with the way to mitigate such an issue.

---

> > ### Comment · Reviewer_4ttG · 2025-11-28
> >
> > Thank you, I will keep my score as is 6

---

### Official Review · Reviewer_TZHK · 2025-10-27

**Soundness:** 2
**Presentation:** 3
**Contribution:** 2
**Rating:** 4
**Confidence:** 4

**Summary:**

This paper proposes KME-CLIP, an enhancement to CLIP's similarity function. The core idea is that the optimal similarity metric, Pointwise Mutual Information (PMI), is not well-approximated by CLIP's standard cosine similarity. The authors' key insight is that under a conditional independence assumption, the exponential of PMI can be expressed as an inner product in an $L^2$ space. To exploit this, KME-CLIP represents image and text embeddings as kernel mean embeddings in a Reproducing Kernel Hilbert Space. Theoretical results and experiments verifies their advantage towards current methods.

**Strengths:**

The paper provides sufficient theoretical proof that KME-CLIP can approximate PMI with arbitrary accuracy. The method theoretically and empirically outperforms standard CLIP and WPSE on different tasks.

**Weaknesses:**

1. Assumption 2 is quite hard to hold for real data distribution. It is highly unlikely that a single latent topic $z$ is sufficient to make a complex image and its detailed textual description completely independent.
2. The motivation of the paper could be problematic. It is built on the current methods' non-linear approximation of $exp(PMI)$, including CLIP's. However, Proposition 2 indicates that the CLIP similarity can be written as a one-point version of the proposed KME-CLIP. This violates the initial motivation. The effectiveness may only come from the introduction of multiple embeddings.
3. The proposed similarity calculation requires $O(m_X m_Y)$ kernel computations for each pair in the $B \times B$ similarity matrix. This leads to higher computational and hardware requirements as shown in Appendix Table 7.

**Questions:**

When switching to WPSE's similarity function in the current framework, how would the model perform?

Please also see Weaknesses.

---

> ### Author Response · Authors · 2025-11-21
>
> We are grateful for your helpful and insightful review comments. We appreciate that you have acknowledged the fact that our method surpasses both CLIP and WPSE from theoretical as well as experimental perspectives.
>
> ---
>
> > Assumption 2 is quite hard to hold for real data distribution. It is highly unlikely that a single latent topic is sufficient to make a complex image and its detailed textual description completely independent.
>
> As is stated in [1, Remark 3.2], this assumption is frequently made in theoretical analysis of contrastive learning and holds if we do not require additional regularity conditions (e.g., simply assigning $z=x$ makes $y|z$ independent from $x|z$ as the latter is deterministic). While it is nontrivial if there exists a “good” latent with conditional independence and a certain degree of regularity in real data, we would argue that working on conditional independence at the stage of theoretical analysis is a reasonable abstraction.
>
> [1] Zixiang Chen, Yihe Deng, Yuanzhi Li, Quanquan Gu, “Understanding Transferable Representation Learning and Zero-shot Transfer in CLIP”, ICLR 2024.
>
> ---
>
> > The motivation of the paper could be problematic. It is built on the current methods' non-linear approximation of $\exp(\text{PMI})$, including CLIP's. However, Proposition 2 indicates that the CLIP similarity can be written as a one-point version of the proposed KME-CLIP. This violates the initial motivation. The effectiveness may only come from the introduction of multiple embeddings.
>
> There might be a misunderstanding regarding our use of the term "linearity". By this term, we specifically refer to the fact that $\exp(\text{PMI})$ exhibits a bi-linear structure, as it can be interpreted as an inner product in an $L^2$ space. In this sense, the similarity function used in CLIP does not possess such a bi-linear structure, because a single term $\exp(f_{\mathcal{X}}(x)^\top f_{\mathcal{Y}}(y))$ is not bi-linear with respect to functions of $x$ and $y$ due to the non-linear nature of the $exp$ transformation.
>
> To illustrate this point concretely, the CLIP similarity cannot represent a midpoint between two points. For example, for some $x_1, x_2 \in \mathcal{X}$, the quantity
> \begin{equation}
> (\exp(f_{\mathcal{X}}(x_1)^\top f_{\mathcal{Y}}(y))+\exp(f_{\mathcal{X}}(x_2)^\top f_{\mathcal{Y}}(y)))/2
> \end{equation}
> cannot, in general, be expressed as $\exp(f_{\mathcal{X}}(x_3)^\top f_{\mathcal{Y}}(y))$ for any $x_3 \in \mathcal{X}$.
>
> In contrast, the similarity of KME-CLIP admits a bi-linear structure as an inner product in an RKHS. Consequently, the similarity in KME-CLIP can indeed represent a midpoint, since
> \begin{equation}
> (\langle h_\theta(x_1),h_\theta(y)\rangle_{\mathcal{H}}+\langle h_\theta(x_2),h_\theta(y)\rangle_{\mathcal{H}})/2=\langle (h_\theta(x_1)+h_\theta(x_2))/2, h_\theta(y)\rangle_{\mathcal{H}}.
> \end{equation}
>
> This distinction arises from the difference in representation capacity within the function space. More specifically, CLIP can be viewed as a special case of KME-CLIP in which the coefficients are fixed as $w_i^{\mathcal{X}}(x)=1$ and $w_i^{\mathcal{Y}}(y)=1$, which results in a more limited representation capacity in the CLIP space.
>
> ---
>
> > The proposed similarity calculation requires $O(m_{\mathcal{X}} m_{\mathcal{Y}})$ kernel computations for each pair in the $B\times B$ similarity matrix. This leads to higher computational and hardware requirements as shown in Appendix Table 7.
>
> As you correctly point out, the computational cost increases when using the full outputs of the Transformer. Therefore, in practical applications, it is important to choose appropriate values for $m_{\mathcal{X}}$ and $m_{\mathcal{Y}}$ by considering the trade-off between computational efficiency and accuracy. We would like to emphasize that, as shown in Table 6 in Appendix A.3, our method requires the same number of GPUs as CLIP as long as the value of $m_{\mathcal{X}}$ is not increased, while still achieving improved performance compared to CLIP.
>
> ---
>
> > When switching to WPSE's similarity function in the current framework, how would the model perform?
>
> We may not be fully certain that we have correctly interpreted the reviewer’s question. Specifically, we are unsure whether the question refers to evaluating the performance of simply applying WPSE as is, or to conducting the same similarity computation within the RKHS induced by WPSE (i.e., using a kernel over an RKHS).
> If the intended meaning is the former, the performance of WPSE is reported in Tables 1 and 2, where our method consistently outperforms it.
> If the intended meaning is the latter, the implementation might be tricky since the model is required to represent a set of point sets, which quickly becomes infeasible if we want precise approximations for the two RKHSs in a recursive relationship.

---

> > ### Comment · Reviewer_TZHK · 2025-11-28
> > **Reply to the authors**
> >
> > The authors have addressed some of my concerns. For the solid theoretical contribution of the paper, I would raise my score to 6. However, I would need stronger evidence confirming that the empirical improvements stem directly from this theoretical advantage.
> >
> > Meanwhile, I strongly suggest the authors emphasize the intuition derived from Theorems 6 & 7 earlier in the text, explicitly contrasting them would much better justify the motivation than the general RKHS discussion alone.

---

> > > ### Author Response · Authors · 2025-12-01
> > >
> > > Thank you for your follow-up comments. Let us reply to them below.
> > >
> > > > The authors have addressed some of my concerns. For the solid theoretical contribution of the paper, I would raise my score to 6.
> > >
> > > Thank you for appreciating our rebuttal and theoretical contributions. We are glad to hear the reviewer's intention for raising score. Let us address the remaining concerns in the following:
> > >
> > > >I would need stronger evidence confirming that the empirical improvements stem directly from this theoretical advantage.
> > >
> > > Although it is difficult to confirm that the the empirical improvements fully stems from the theoretical advantages, we can partially understand the implications of our theory through the comparison between WPSE and KME-CLIP. While WPSE and our method share the common property of utilizing RKHS vectors, and their loss functions are quite similar (differing primarily in the logarithmic operation), their empirical behaviors differ substantially, as shown in Tables 1 and 2. Their difference gets even more highlighted when we look at the ablation studies in the WPSE paper, as we explained in the final reply to Reviewer 3Vp7, summarized as follows:
> > >
> > > - In WPSE, using linear kernels (that is theoretically equivalent to CLIP except for the normalization) rather than nonlinear kernels does not change the zero-shot performance in their ablation. In contrast, KME-CLIP can clearly separate it from CLIP in theory (Theorems 6, 7) and its empirical performance.
> > > - Enforcing positive weights in WPSE leads to corrupting behavior, whereas the use of positive weights in KME-CLIP is not at all problematic, as is predicted by theory.
> > >
> > > Finally, as discussed in Section 3.4.2, a major theoretical distinction between our method and WPSE lies in their respective abilities to approximate PMI: WPSE faces difficulties in approximating PMI between two non co-ocurring concepts, whereas our method is able to approximate it effectively. We believe this observation, together with the above comparison, strengthens the evidence that our method's theoretical advantages actually contribute to the empirical improvements.
> > >
> > > >Meanwhile, I strongly suggest the authors emphasize the intuition derived from Theorems 6 & 7 earlier in the text, explicitly contrasting them would much better justify the motivation than the general RKHS discussion alone.
> > >
> > > Thank you for the suggestion. We have revised Section 3.4.1 by adding an explanation of the essential differences between CLIP and our method, and we have also incorporated references to Theorems 6 and 7.

---

### Official Review · Reviewer_MYi2 · 2025-11-01

**Soundness:** 3
**Presentation:** 3
**Contribution:** 3
**Rating:** 6
**Confidence:** 3

**Summary:**

This paper introduces KME-CLIP, a novel method that refines the CLIP framework by theoretically identifying and exploiting the linear structure of the exponential Pointwise Mutual Information (exp(PMI)) in a Reproducing Kernel Hilbert Space (RKHS). The authors provide theoretical guarantees on the approximation capability of their method and demonstrate consistent, albeit sometimes modest, empirical improvements over baselines across a range of standard multimodal tasks.

**Strengths:**

1. The paper presents a well-formulated and mathematically grounded connection between PMI, RKHS, and contrastive learning objectives.
2. The authors clearly delineate differences and relationships between CLIP, WPSE, and their proposed model.
3. KME-CLIP shows small but stable improvements in retrieval and zero-shot classification across a variety of datasets.
4. The exposition is clear and professional.

**Weaknesses:**

1. The empirical methodology closely follows existing work (especially WPSE), with only a minor modification in similarity computation.
2. The observed performance gains are small (typically 1–3%), raising questions about practical significance relative to computational overhead.
3. The paper exclusively uses Gaussian kernels without justifying why other positive definite kernels (e.g., Laplacian) are not considered, nor exploring their impact on performance.
4. The paper lacks a detailed evaluation of training/inference cost or memory footprint, which is critical for kernel-based approaches.

**Questions:**

1. Why is the linear classification performance not improved despite a superior PMI approximation? Could post-RKHS projection features or modified feature extraction pipelines address this gap?
2. How sensitive is the method to the choice of kernel and its parameters? Are there scenarios where alternative kernels outperform Gaussian kernels?
3. Can KME-CLIP be efficiently applied to large-scale pretrained CLIP models without full retraining?
4. For practical deployment, what is the recommended point set size to balance performance gains and inference cost? Is there a heuristic to determine this size for different datasets/tasks?
5. Does the RKHS similarity mitigate or exacerbate the “modality gap” observed in prior CLIP analyses?
6. Could the RKHS log-inner-product formulation be combined with in-modal self-supervision (e.g., as in SLIP)?

---

> ### Author Response · Authors · 2025-11-21
>
> Thank you for your valuable and helpful comments. We appreciate that you have recognized both the theoretical analysis underpinning our approach and the improvements it brings in practical performance.
>
> ---
>
> > The empirical methodology closely follows existing work (especially WPSE), with only a minor modification in similarity computation.
>
> While the form of our loss function is similar to that of WPSE, the underlying mathematical structure of our approach is fundamentally different. WPSE is motivated by transforming CLIP embeddings into infinite-dimensional vectors from the perspective of representational capacity, whereas our approach aims at leveraging inherent linear structure in exp(PMI), which is ignored both by CLIP and WPSE. Consequently, as discussed in Section 4 (and is acknowledged by the review), the theoretical properties of KME-CLIP differ substantially from those of WPSE, and the experimental results in Tables 1 and 2 demonstrate that KME-CLIP achieves better performance than WPSE.
> We have also added further clarification on these differences in Section 3.4.2 in the revised manuscript.
>
> ---
>
> > The paper exclusively uses Gaussian kernels without justifying why other positive definite kernels (e.g., Laplacian) are not considered, nor exploring their impact on performance.
>
> We use Gaussian kernels for three reasons, actually. First is the positivity of the value. The positivity is needed since we take the logarithm for the inner product. Second is the capability of taking the value near zero. This is for approximating a situation where $p(x,y)=0$ holds. Such situations occur when the image and provided caption do not correspond at all, an image of a dog and a text “this is a cat” for instance. Third is since by Gaussian kernels, our method involves CLIP as a special case. This property allows us to anticipate that our method will, at the very least, outperform CLIP.
> To clarify these points, we have added justifications for using Gaussian kernels in Section 3.3.
>
> ---
>
> > The paper lacks a detailed evaluation of training/inference cost or memory footprint, which is critical for kernel-based approaches.
>
> The required computational cost is reported in Appendix A.3. As shown in Table 7, the number of GPUs needed for training increases as the point set becomes larger. However, when we reduce the size of point set to 10, the required number of GPUs becomes the same as that of CLIP (Table 6), while still achieving better performance than CLIP (Table 3).
>
> ---
>
> > Why is the linear classification performance not improved despite a superior PMI approximation? Could post-RKHS projection features or modified feature extraction pipelines address this gap?
>
> As the reviewer correctly points out, since the linearity induced by the RKHS projection is not utilized in linear classification, this setting is not directly comparable to our proposed method. This is likely the primary reason for the lack of performance improvement.
>
> ---
>
> > How sensitive is the method to the choice of kernel and its parameters? Are there scenarios where alternative kernels outperform Gaussian kernels?
>
> We learn the parameters of the kernel. The reason for using Gaussian kernels are written above.
>
> ---
>
> > Can KME-CLIP be efficiently applied to large-scale pretrained CLIP models without full retraining?
>
> As the reviewer correctly notes, because the similarity function of CLIP can be viewed as a one-point instance of KME-CLIP, it is in principle possible to use a pretrained CLIP model as an initialization for training KME-CLIP.
> Since, in our implementation, the position of the ViT output used by CLIP varies depending on the input data, this training strategy is not directly applicable to our framework. However, it remains a promising direction for future work, provided that such implementation details are fixed.
>
> ---
>
> > For practical deployment, what is the recommended point set size to balance performance gains and inference cost? Is there a heuristic to determine this size for different datasets/tasks?
>
> We simply recommend using the largest point-set size that the available computational resources allow. The primary bottleneck is memory consumption, and the required memory scales as $O(m_X m_Y)$ relative to that of CLIP.
>
> ---
>
> > Does the RKHS similarity mitigate or exacerbate the “modality gap” observed in prior CLIP analyses?
>
> Theoretically, as stated in Theorem 7, KME-CLIP can approximate $exp(PMI)$ while maintaining a low modality gap in a toy setting.
>
> ---
>
> > Could the RKHS log-inner-product formulation be combined with in-modal self-supervision (e.g., as in SLIP)?
>
> Combining our method with self-supervised learning by adding a self-supervised loss to the KME-CLIP objective is an interesting direction. Indeed, our theory in Section 4.3 is based on the fact that KME-CLIP can approximate both in-modal and cross-modal (i.e., usual) PMIs at the same time. We thus anticipate they can be combined, while further investigation is needed.

---

### Official Review · Reviewer_3Vp7 · 2025-11-03

**Soundness:** 3
**Presentation:** 3
**Contribution:** 2
**Rating:** 2
**Confidence:** 4

**Summary:**

The paper proposes to replace the point-wise contrastive loss of CLIP with a dense contrastive loss calculated using the idea of PMI.

The authors show that this leads to better results (using the same evaluation as CLIP).

The method is rigorously motivated by theoretical analysis.

**Strengths:**

- The paper is well presented and the idea is valid.
- Figure 1 is a clear illustration of the method. Using the intermediate features of the encoders rather than just the output features is innovative.
- The authors include a Section 3.4 comparing their method with the baselines. This is helpful for readers to compare with existing literature.

**Weaknesses:**

The main issue is that this paper is substantially similar to [Uesaka et al. ICLR 2025]
 - The authors acknowledge this in Section 3.4.2. The distinctions between the current work and [Uesaka et al. ICLR 2025] are minimal.
 - Please compare Figure 1 of the two papers - I think the idea being presented is identical, with minor differences in implementation.
 - The theoretical contributions are different.

I am ok with acceptance if the above concern is not valid after discussion.

Secondary concerns:
- The paper only compares against CLIP and WPSE. There have been many improvements to the CLIP loss. [Dong], for example.
   - While these other variants of CLIP do not try to approximate PMI, they should be used as baselines.


[Dong] Don et al. CVPR 2023. MaskCLIP: Masked Self-Distillation Advances ContrastiveLanguage-Image Pretraining.

**Questions:**

- In the introduction, what do you mean by " It is theoretically known that the similarity function optimal in terms of the contrastive loss corresponds to the pointwise mutual information (PMI), defined as log(p(x, y)/(p(x)p(y))) for two datapoints x and y from different modalities (Oord et al., 2018; Zhang et al., 2023)."?
How do we defined the optimal similarity function here?
I think more discussion about why we want to use PMI instead of other contrastive losses would be helpful before delving into the details of how to optimize PMI better. Thank you.

---

> ### Author Response · Authors · 2025-11-21
>
> Thank you for a fruitful and insightful review. We appreciate that you have acknowledged both the theoretical insights that motivate our approach and its improved performance over CLIP.
>
> ---
>
> > The main issue is that this paper is substantially similar to [Uesaka et al. ICLR 2025].
> The authors acknowledge this in Section 3.4.2. The distinctions between the current work and [Uesaka et al. ICLR 2025] are minimal.
> Please compare Figure 1 of the two papers - I think the idea being presented is identical, with minor differences in implementation.
> The theoretical contributions are different.
> I am ok with acceptance if the above concern is not valid after discussion.
>
> While the form of our loss function is indeed similar to that of WPSE, the underlying mathematical structure of our approach is fundamentally different. WPSE is motivated by transforming CLIP embeddings into infinite-dimensional vectors from the perspective of representational capacity, whereas our approach aims at leveraging inherent linear structure in exp(PMI), which is ignored both by CLIP and WPSE. Consequently, as discussed in Section 4 (and is acknowledged by the review), the theoretical properties of KME-CLIP differ substantially from those of WPSE, and the experimental results in Tables 1 and 2 demonstrate that KME-CLIP achieves better performance than WPSE.
> We have also added further clarification on these differences in Section 3.4.2 in the revised manuscript.
>
> ---
>
> > The paper only compares against CLIP and WPSE. There have been many improvements to the CLIP loss. [Dong], for example.
> While these other variants of CLIP do not try to approximate PMI, they should be used as baselines.
> [Dong] Don et al. CVPR 2023. MaskCLIP: Masked Self-Distillation Advances ContrastiveLanguage-Image Pretraining.
>
> As you correctly point out, there have been numerous improvements to the CLIP loss, including both your proposed method and the approaches listed in our Related Work section. However, conducting empirical comparison against all related methods within a limited period is not feasible. We believe that the comparisons with CLIP and WPSE sufficiently support the main claims of our work, especially within the methods aiming at PMI approximations. Nevertheless, we appreciate your valuable suggestion and have added the referenced paper to the Related Work section.
>
> ---
>
> > In the introduction, what do you mean by " It is theoretically known that the similarity function optimal in terms of the contrastive loss corresponds to the pointwise mutual information (PMI), defined as log(p(x, y)/(p(x)p(y))) for two datapoints x and y from different modalities (Oord et al., 2018; Zhang et al., 2023)."? How do we defined the optimal similarity function here? I think more discussion about why we want to use PMI instead of other contrastive losses would be helpful before delving into the details of how to optimize PMI better. Thank you.
>
> This sentence indicates that if the similarity function corresponds to PMI, then the population contrastive loss is minimized with respect to the choice of similarity function. We consider minimizing the population loss to be an important research direction, as it reflects the average accuracy of image-to-text and text-to-image retrievals and is supported by theoretical guarantees [1]. Although exploring alternative contrastive losses is indeed a valuable direction for future work, as you point out, our paper focuses on addressing the mismatch present in the current contrastive loss used in CLIP. We have added additional explanation regarding the motivation for using PMI in Section 2.2 of the revised manuscript.
>
> [1] Kazusato Oko, Licong Lin, Yuhang Cai, and Song Mei. A statistical theory of contrastive pretraining and multimodal generative ai. arXiv preprint arXiv:2501.04641, 2025.

---

> > ### Comment · Reviewer_3Vp7 · 2025-11-28
> >
> > My main concern regarding similarity to WPSE remains unaddressed. This concern is shared by two other reviewers. I understand the authors are framing the paper such that the mathematical motivation is different, but if the form of the resulting optimization objective is similar, the impact may be limited.
> >
> > Additionally, since the form of the optimization objective is so similar to WPSE, it remains unclear where the empirical gains are coming from, is it because the authors put more effort into optimizing the hyperparameters of their proposed method, or is there some more robust explanation? Is it just because of the extra log? I think reviewer TZHK's question "When switching to WPSE's similarity function in the current framework, how would the model perform?" is in this same direction.

---

> > > ### Author Response · Authors · 2025-11-28
> > >
> > > Thank you for your follow-up comment. We believe we now understand the reviewer's concerns more clearly, so let us address them.
> > >
> > > > the form of the resulting optimization objective is similar
> > >
> > > > Is it just because of the extra log?
> > >
> > > To these points, we would say that the extra log indeed plays a crucial role, since it determines a completely different objective. We can understand it by looking at the ablation studies of the WPSE paper [1]. **In this answer, the table number refers to that of their paper.**
> > >
> > > **WPSE needs a linear kernel in addition to Gaussian:** While the framework of WPSE is applicable to any positive definite kernel, their proposed choice for `WPSE` is a weighted sum of a nonlinear kernel (Guassian or IMQ) and a linear kernel $k(x, y)=x^\top y$ (their preferred choice of coefficients are $1/3$ for nonlinear and $2/3$ for linear). Thus, in Table 6 of [1], they compare WPSE with `WPSE Nonlinear` and `WPSE Linear`, solely using nonlinear or linear kernels. To add context, `WPSE Linear` is mathematically same as CLIP in that the embedding in the end is just a 512-dimensional vector (while the normalization is slightly different).
> > >
> > >  In general, `WPSE Nonlinear` performs worst, and solely using the linear kernel (`WPSE Linear`) almost yields the same zero-shot performance as WPSE in their Table 6, as is also observed in Table 3. This shows that the zero-shot performance of WPSE is not boosted by the introduction of RKHS, but rather by computing the CLIP embedding by linearly aggregating the transformer outputs. In contrast, WPSE constently outperforms `WPSE Linear` in linear classification (in Tables 3, 6), which alignes with their theoretical discussion around the linear classification (e.g., their Theorem 4.4). In that they require linear kernels to catch up with the CLIP-like embedding, WPSE is fundamentally different from KME-CLIP, which outperforms CLIP/WPSE in retreival and zero-shot tasks with a single Gassian kernel.
> > >
> > > **WPSE with positive weights does not perform well:** In Table 3 of [1], the authors of WPSE conduct ablation studies using positive weights (though it is with IMQ kernel). In the experiments, training WPSE with positive weights does not perform well (at all). Thus, negative weights is crucial in the performance of WPSE. In constrast, KME-CLIP just uses positive weights for exploiting the linear structure of $\exp(\text{PMI})$.
> > >
> > > As discussed in Remark 1 of our paper, WPSE needs to approximate negative values (which can diverge in the direction of $-\infty$) by the scaled inner product of $\langle u, v\rangle_{\mathcal{H}}/\tau$ (please see also the top of our Section 3.4.2). This leads to the requirement of negative innder product values and the reason for the failure of WPSE with positive weights. In contrast, KME-CLIP can represent the case with $p(x,y) \to 0$ and $p(x),p(y)>0$ just by placing the corresponding point sets a bit apart, thanks to the decay of Gaussian. We would argue that enforcing negative inner products (as required in CLIP and WPSE) for each non co-occurring pair of events can take extra effort than just locating them in different places (as in KME-CLIP), because the former further requires that the embeddings are in opposite directions.
> > >
> > > >  it remains unclear where the empirical gains are coming from, is it because the authors put more effort into optimizing the hyperparameters of their proposed method, or is there some more robust explanation?
> > >
> > > For this comment, we believe our explanation above gives a robust explanation. Regarding the optimization, we actually did not conduct any hyperparameter search and just used those given in the WPSE's codebase. The only exeption is the initial value of learnable lengthscale $1/\sigma=3.779645$ (`sigma_inv` in the code), which is set to align with the initial logit scale $\tau=14.2857$ of WPSE under the equation $\tau=1/\sigma^2$. We noticed that we inadvertently forgot to include this KME-CLIP-specific intial value in the paper (whereas the equation $\tau=1/\sigma^2$ is written at the end of "Implementation of KME-CLIP" in Section A.1), so we will add this information in the revision. Thank you for letting us notice that.
> > >
> > > In conclusion, we claim that KME-CLIP and WPSE are fundamentally different despite their apparent similairy in the objective funciton. While we believe this fact is well-supported by the theoretical differences, we can also confirm the differences via the ablation studies in WPSE as discussed above. Please let us know if there remains any concerns in this regard.
> > >
> > >
> > > [1] Uesaka et al. Weighted Point Set Embedding for Multimodal Contrastive Learning Toward Optimal Similarity Metric. ICLR 2025.

---

### Official Review · Reviewer_eh24 · 2025-11-04

**Soundness:** 4
**Presentation:** 3
**Contribution:** 3
**Rating:** 8
**Confidence:** 3

**Summary:**

This paper derives a theory-backed similarity metric for comparing two modalities in CLIP-type models. The main idea is to use approximate pointwise mutual information (PMI), which is implemented via kernel mean embeddings utilizing the linear structure of exp(PMI). The results show consistent improvement over similar CLIP-type methods.

**Strengths:**

1. The derivation is neat and backed by theory. The insight of the linear structure of PMI with the conditional independence assumption makes sense, and this seems to make a major difference in practice -- comparison against WPSE.
2. Significant and consistent improvements against WPSE and CLIP in various retrieval and zero-shot classification tasks.
3. Writing is clear, and the arguments are adequately motivated.

**Weaknesses:**

1. The low modality gap assumption in Section 4.2 may not be realistic [1]. It would be better to clarify why this assumption is needed and what happens in scenarios where this is violated.
2. Related to this, would the proposed method yield a lower modality gap?
3. How does the choice of kernel function affect the results or approximation quality?

[1] Liang, Victor Weixin, et al. "Mind the gap: Understanding the modality gap in multi-modal contrastive representation learning." Advances in Neural Information Processing Systems 35 (2022): 17612-17625.

**Questions:**

1. The proposed method does not seem specific to the hypersphere. Could other geometric spaces be used to obtain features?
2. Would it make sense to consider this similarity metric under the paradigm of accepting the modality gap similar to [2]?

[2] Ramasinghe, Sameera, et al. "Accept the modality gap: An exploration in the hyperbolic space." Proceedings of the IEEE/CVF Conference on Computer Vision and Pattern Recognition. 2024.

---

> ### Author Response · Authors · 2025-11-21
>
> Thank you for a fruitful and insightful review. We appreciate that you have recognized both the theoretical grounding of our method and its practical effectiveness.
>
> ---
>
> > The low modality gap assumption in Section 4.2 may not be realistic [1]. It would be better to clarify why this assumption is needed and what happens in scenarios where this is violated. \
> [1] Liang, Victor Weixin, et al. "Mind the gap: Understanding the modality gap in multi-modal contrastive representation learning." Advances in Neural Information Processing Systems 35 (2022): 17612-17625.
>
> > Related to this, would the proposed method yield a lower modality gap?
>
> Thank you for the comment. Actually, our intention is to provide a theoretical insight that there are cases where satisfying “low-modality gap” and “small contrastive loss” at the same time is difficult for CLIP, and KME-CLIP does not suffer from such an issue. To clarify this context, we have revised our statements in the paper (written in blue). While mathematically equivalent to the previous statements, the revised ones would convey our intention more directly.
>
> ---
>
> > How does the choice of kernel function affect the results or approximation quality?
>
> We use Gaussian kernels for three reasons, actually. First is the positivity of the value. The positivity is needed since we take the logarithm for the inner product. Second is the capability of taking the value near zero. This is for approximating a situation where $p(x,y)=0$ holds. Such situations occur when the image and provided caption do not correspond at all, an image of a dog and a text “this is a cat” for instance. Third is since by Gaussian kernels,  our method involves CLIP as a special case. This property allows us to anticipate that our method will, at the very least, outperform CLIP.
> To clarify these points, we have added justifications for using Gaussian kernels in Section 3.3.
>
> ---
>
> > The proposed method does not seem specific to the hypersphere. Could other geometric spaces be used to obtain features?
>
> As the reviewer correctly points out, our method can, in principle, be applied to geometric spaces other than the hypersphere. Although our implementation uses a hyperspherical embedding space, the approximation properties established in Theorems 3, 4, and 5 can be applied to a general class of embedding spaces associated with Euclidean distances, which would be an interesting direction for future work. We sticked to the hypersphere formulation so that KME-CLIP works as a generalization of CLIP, as stated in Proposition 2.
>
> ---
>
> > Would it make sense to consider this similarity metric under the paradigm of accepting the modality gap similar to [2]? \
> [2] Ramasinghe, Sameera, et al. "Accept the modality gap: An exploration in the hyperbolic space." Proceedings of the IEEE/CVF Conference on Computer Vision and Pattern Recognition. 2024.
>
> We appreciate the reviewer for bringing this interesting paper to our attention, and we agree that this research direction is highly intriguing. Indeed, Eq. (7) in the reference suggests that it is based on a Gaussian kernel with the geodesic distance. We could naturally extend it by considering its linear combinations as done in CLIP to KME-CLIP. However, the theory part including the use of RKHS would require nontrivial modifications since the resulting kernel is not readily positive definite (cf: Schoenberg’s theorem), so we need to carefully investigate such generalizations.
>
> ---

---

### Comment · Area_Chair_yNAJ · 2025-11-25
**Post-Rebuttal Discussion**

Dear all,

Could you first review the original comments from other reviewers and the rebuttal materials, and then post your comments? Discussion is necessary for this paper.

Best,
AC

---

### Author Response · Authors · 2025-11-26

Dear Reviewers,

We sincerely appreciate your thoughtful and insightful feedback.
In response to your comments, we have revised the manuscript as summarized below.
All revised or newly added sentences are highlighted in blue in the updated version.
Please let us know if our rebuttal and revision address the reviewers' concerns.

---

**Difference from WPSE [Uesaka et al, 2025]**
- We have expanded Section 3.4.2 to clarify the differences in motivation and theoretical properties between our method and WPSE.

**The reason for the choice of Gaussian kernel**
- We have added an explanation in Section 3.3 detailing why we primarily adopt the Gaussian kernel as the kernel function for KME-CLIP.

**The motivation for approximating PMI**
- We have augmented Section 2.2, immediately below Theorem 1, to explain why approximating PMI is an important research direction.

**The theoretical claims in section 4.2**
- We have revised the statements of Theorems 6 and 7 to improve readability and reduce potential ambiguity.

**Related work**
- We have included additional related work in Section 6.

**Other minor revisions**
- We have refined several expressions throughout the manuscript, including those in the introduction, to improve overall readability.

---

### Author Response · Authors · 2025-12-03
**Summary of discussions**

Dear Area Chair(s),

As our discussion phase was closed before reaching conclusion, let us summarize the remaining points after the partial discussions with Reviewers. Three reviewers are positive about our paper, and the two remaining reviewers share related concerns after our rebuttal:

- **Similarity to WPSE** (Reviewer 3Vp7): The reviewer constructively stated "I am ok with acceptance if the above concern is not valid after discussion" in the initial review, referring to the similarity to WPSE [1]. While the reviewer acknowledged that our theoretical contributions are different from WPSE, they still pointed out that the objective functions of KME-CLIP and WPSE are very similar, and thus the impact may be limited. In response, we highlighted differences revealed by the ablation studies conducted in [1, Tables 3, 6] and argued that their distinction beyond theory is apparent in WPSE's ablation studies as follows:
    - WPSE with a single nonlinear kernel does not perform well even compared to linear kernel, while KME-CLIP with Gaussian can outperform them without additional hyperparameter search.
    - WPSE with positive weights leads to corrupted performance, while the positivity of the weights is the core of KME-CLIP's theory.

    Please refer to our final reply to Reviewer 3Vp7 for more detailed discussions.

- **Need stronger evidence for theory directly leading to empirical improvements** (Reviewer TZHK): In the follow-up reply to our rebuttal, the reviewer stated "For the solid theoretical contribution of the paper, I would raise my score to 6," and the remaining concern was if the empirical improvements directly stem from theory. While it is difficult to fully confirm it, we argued in reply that the ablation studies in WPSE (mentioned above) strengthen the evidence.  Please refer to our final reply to Reviewer TZHK for more detailed discussions.

Reference:

[1] Uesaka et al. Weighted Point Set Embedding for Multimodal Contrastive Learning Toward Optimal Similarity Metric. ICLR 2025.

---

### Meta-Review · Area_Chair_MiEJ · 2026-01-05

**Summary:**

The paper proposes KME-CLIP, a theoretical refinement to the similarity computation in CLIP. The authors identify that CLIP fails to fully utilize the linear structure of pointwise mutual information (PMI). They propose mapping embeddings into a Reproducing Kernel Hilbert Space (RKHS) where the inner product can approximate $exp(PMI)$ with arbitrary accuracy. The empirical results showed effectiveness over CLIP and WPSE across various tasks.

**Reviewer Concerns:**

Reviewer 3Vp7 (and to a lesser extent MYi2) argued that the resulting optimization objective is substantially similar to WPSE. He/she questions whether the empirical gains stem from the new theory or minor implementation details/hyperparameters. The authors provided a detailed rebuttal highlighting fundamental theoretical differences. Specifically, KME-CLIP utilizes positive weights to exploit the linear structure of $exp(PMI)$, whereas WPSE relies on negative weights to approximate values, a distinction supported by WPSE's own ablation studies.

**Reviewer Scores:**

In their final comment, This reviewer 3Vp7  explicitly stated, "My main concern regarding similarity to WPSE remains unaddressed... I think the idea being presented is identical." The reviewer MYi2 acknowledged the strengths but remained concerned about the small empirical gains and methodological similarities to existing work.

---

### Decision · Program_Chairs · 2026-01-26

Reject